# To Grok Grokking: Provable Grokking in Ridge Regression

**Mingyue Xu**[1]  **Gal Vardi**[2]  **Itay Safran**[3]

## Abstract

We study *grokking*—the onset of generalization long after overfitting—in a classical ridge regression setting. We prove end-to-end grokking results for learning over-parameterized linear regression models using gradient descent with weight decay. Specifically, we prove that the following stages occur: (i) the model overfits the training data early during training; (ii) poor generalization persists long after overfitting has manifested; and (iii) the generalization error eventually becomes arbitrarily small. Moreover, we show, both theoretically and empirically, that grokking can be amplified or eliminated in a principled manner through proper hyperparameter tuning. To the best of our knowledge, these are the first rigorous quantitative bounds on the generalization delay (which we refer to as the "grokking time") in terms of training hyperparameters. Lastly, going beyond the linear setting, we empirically demonstrate that our quantitative bounds also capture the behavior of grokking on non-linear neural networks. Our results suggest that grokking is not an inherent failure mode of deep learning, but rather a consequence of specific training conditions, and thus does not require fundamental changes to the model architecture or learning algorithm to avoid.

## 1. Introduction

By the standard of classical machine learning, overfitting the training data is usually believed to be harmful to generalization. To overcome this, regularization techniques such as early-stopping, dropout and weight decay have been developed. However, deep learning can sometimes exhibit

counterintuitive phenomena that contravene this traditional wisdom. One of the most striking examples is "grokking" (Power et al., 2022), a phenomenon where generalization only starts improving long after the model achieves perfect training performance. The phenomenon of grokking has been extensively studied in recent years (Chughtai et al., 2023; Tan & Huang, 2023; Notsawo Jr et al., 2023; Fan et al., 2024), and was also encountered in models that are different from neural networks (Blanc et al., 2020; Humayun et al., 2024; Merrill et al., 2023), as well as in large language models (Zhu et al., 2024; Wang et al., 2024; Xu et al., 2025) (see Subsection 1.1 for further discussion). But despite its increasing popularity, only a few prior works have established rigorous theoretical guarantees.

Several existing theoretical papers attribute the occurrence of grokking to a transition in the optimization dynamics from the lazy to the rich regime (Lyu et al., 2024; Mohamadi et al., 2024; Kumar et al., 2024). Among these works, Lyu et al. (2024) consider training homogeneous neural networks for both classification and regression problems using gradient flow with weight decay, and prove a sharp transition between the kernel and rich regimes. However, their technique only guarantees convergence to a KKT (Karush-Kuhn-Tucker) point, which is, in general, not sufficient for arguing global optimality, and their result does not provably imply grokking. In another work, Mohamadi et al. (2024) provide a theoretical foundation for grokking on the specific regression problem of modular addition. They consider training two-layer quadratic networks with an $\ell_\infty$ regularization, and show that grokking is a consequence of the transition from kernel-like behavior to the limiting behavior of gradient descent (GD). Yet, their analysis does not establish that GD converges to a solution with a small weight norm that generalizes well, despite empirically verifying this. To the best of our knowledge, the work closest to showing a provable grokking result is Xu et al. (2024). By studying a high-dimensional clustered XOR dataset in a binary classification setting, they show that after a catastrophic one-step overfitting—where generalization is no better than a random guess—the model eventually achieves perfect test accuracy. Still, they do not show that the onset of generalization is delayed beyond the first iteration of GD, which could begin as early as the second iteration. While many prior works attribute grokking to the transition

[1]Department of Computer Science, Purdue University, West Lafayette, IN, USA [2]Department of Computer Science and Applied Mathematics, Weizmann Institute of Science, Israel [3]Stein Faculty of Computer and Information Science, Ben-Gurion University of the Negev, Israel. Correspondence to: Mingyue Xu <xu1864@purdue.edu>.

*Proceedings of the 43rd International Conference on Machine Learning*, Seoul, South Korea. PMLR 306, 2026. Copyright 2026 by the author(s).

between the lazy and rich regimes during optimization, a recent work by Boursier et al. (2025) focused specifically on studying the role of weight decay in grokking, revealing that grokking can also arise from the transition between the "ridgeless" to the "ridge" regimes. They considered training using gradient flow with weight decay on a smooth loss objective and showed a two-phase behavior of the trajectory. Nevertheless, they do not prove that the unregularized solution does not generalize and the ridge solution does generalize, and thus their results do not imply provable grokking. Relatedly, Tikeng Notsawo et al. (2025) study a general loss-plus-regularizer framework, showing an early loss-minimization phase followed by a regularization-driven phase on a time scale proportional to $1/(\eta\lambda)$. Our work provides sharper end-to-end guarantees in the specific setting of ridge regression.

Given the current theoretical gaps around grokking, our goal is to prove an end-to-end grokking guarantee demonstrating that: (i) the model overfits early, while test performance remains poor; (ii) poor generalization persists long after overfitting; and (iii) the optimization algorithm eventually converges to a model that achieves good generalization. See Figure 1 for an illustration. To this end, we investigate the phenomenon of grokking within a classical teacher-student framework. We specifically consider linear regression, a classical statistical method that has been extensively studied over the centuries (Galton, 1886; Tikhonov, 1963; Hoerl & Kennard, 1970; Uyanık & Güler, 2013). To address overfitting, fundamental regularization techniques such as Lasso and ridge regression have been developed and proven invaluable. Despite being a special case of overfitting, it is surprising that almost no previous work has focused on grokking in linear regression. While prior work has observed this phenomenon mainly in complex, non-linear architectures, our findings reveal that neither deep nor non-linear structural components are strictly necessary for grokking to occur, offering a rigorous, foundational framework for the phenomenon in purely linear settings. To our best knowledge, the only theoretical work on this topic is Levi et al. (2024), in which the authors analyze grokking in a linear regression setting by leveraging random matrix theory. Yet, despite the similar setting being studied, their analysis lacks formal guarantees. Apart from the fact that the results are non-rigorous, their work differs from ours in the following aspects. (1) They assume only Gaussian data, whereas our data distribution is more general; (2) their results consider the regimes where the feature dimension roughly equals the sample size, while we allow arbitrary over-parameterization; and (3) they do not include weight decay as we do.

In this paper, we consider an over-parameterized linear regression problem of learning a ground-truth realizable teacher function, by training a student linear model using randomly initialized GD, on a squared loss objective, and

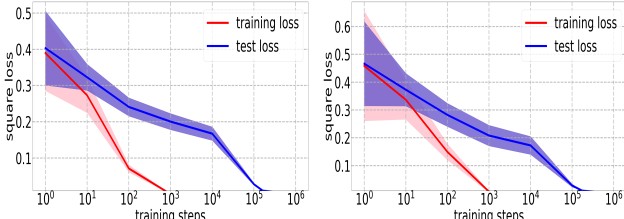

*Figure 1.* Comparing training and test squared losses using GD with weight decay over 50 independent runs (with independent datasets and student initializations). **Left:** training a ridge regression model to learn the zero teacher; **Right:** training a two-layer ReLU neural network (training both layers) to learn the zero teacher. We scale the x-axis logarithmically, which is commonly adopted for plotting the grokking phenomenon (Power et al., 2022; Lyu et al., 2024; Xu et al., 2024; Mohamadi et al., 2024). See Section 5 for further details of the experimental setup.

with an $\ell_2$ regularization. We provide the first end-to-end provable grokking result, and unlike Xu et al. (2024), we are able to prove that poor generalization persists beyond the point where we overfit. Our results hold for a family of teacher functions: any function that is realizable by a linear model over any fixed feature map. Moreover, we also demonstrate, both theoretically and empirically, that grokking can be fully and quantitatively controlled by hyperparameter tuning. Specifically, one way to effectively manipulate the delay before generalization begins is to tune the weight decay, which aligns with the empirical findings of Lyu et al. (2024). To summarize the main novelty in this work, for the first time, (1) we show an end-to-end provable grokking result, while all prior works on grokking in neural networks or linear models did not establish such a comprehensive result; (2) we give a formal analysis of grokking in the fundamental setting of linear/ridge regression; and (3), we provide exact quantitative bounds on the grokking time that reveal how each hyperparameter affects grokking.

In a bit more detail, our main contributions are as follows:

- We study an over-parameterized ridge regression problem (Section 4), and show the first end-to-end provable grokking result (Theorem 4.2) for learning any realizable teacher function using gradient descent. Specifically, we prove the following separate components:
  - We prove that when optimizing the $\ell_2$ regularized squared loss using gradient descent, the training (empirical) squared error decays at a fast convergence rate (Theorem 4.4).
  - We provide a lower bound for the generalization error which admits a slower convergence rate than the one for the training error, implying long-term overfitting (Theorem 4.5).
  - Lastly, we show that gradient descent will eventually reach a global minimum with a generalization guarantee (Theorem 4.6).

- We quantitatively state the sufficient conditions of the hyperparameters to realize grokking (Equations (3), (4) and (5)) and provide a rigorous lower bound on the grokking time in terms of model hyperparameters (Equations (6) and (7)). We then analyze how different hyperparameters affect grokking, which is in keeping with our experimental verifications.
- We support our theoretical findings with empirical simulations that illustrate how grokking can be controlled in a principled manner (Section 5).
- Finally, we conduct experiments on non-linear neural networks, and demonstrate empirically that the dependencies of the grokking time on hyperparameters qualitatively match our provable predictions in linear regression.

## 1.1. Additional Related Work

The intriguing phenomenon of grokking was first coined by Power et al. (2022), who observed it when training on algorithmic datasets for modular addition. However, the underlying behavior—where generalization emerges long after overfitting—had already been noted in earlier work by Blanc et al. (2020), in the context of matrix sensing and simplified shallow neural networks. Following these seminal works, grokking has also been observed in other learning paradigms, which include learning sparse parities (Barak et al., 2022; Merrill et al., 2023), learning the greatest common divisor of integers (Charton, 2024), and image classification (Radhakrishnan et al., 2022).

Several works proposed thoughtful ideas to explain grokking, although none of them provided rigorous end-to-end grokking results. Liu et al. (2022) offered an explanation via the lens of representation learning. Thilak et al. (2022) attributed grokking to the use of adaptive optimizers—an optimization anomaly named the "Slingshot Mechanism" (cyclic phase transitions). Liu et al. (2023) identified a mismatch between training and test loss landscapes, which they called the "LU mechanism", and used it to explain certain aspects of grokking. Nanda et al. (2023) revealed that grokking can arise from the gradual amplification of structured mechanisms encoded in the weights via progress measuring. Davies et al. (2023) hypothesized that grokking and double descent can be understood using the same pattern-learning model, and provided the first demonstration of model-wise grokking. Merrill et al. (2023) studied the problem of training two-layer neural networks on a sparse parity task, and attributed grokking therein to the competition between dense and sparse subnetworks. Gromov (2023) demonstrated that fully-connected two-layer networks exhibit grokking on modular arithmetic tasks without any regularization and attributed grokking to feature learning. Miller et al. (2024) studied grokking beyond neural networks, e.g. Bayesian models. Beck et al. (2025) ex-

plored the phenomenon of grokking in a logistic regression problem. They provided evidence that grokking may occur under some asymptotic assumptions on the hyperparameters, but they did not rigorously prove grokking. Varma et al. (2023) interpreted grokking from the perspective of circuit efficiency, and discovered two related phenomena named "ungrokking" and "semi-grokking". Jeffares et al. (2024) provided empirical insights into grokking, by investigating a telescoping model that suggests that grokking may reflect a transition to a measurably benign overfitting regime during training. Humayun et al. (2024) attributed grokking of deep neural networks to their delayed robustness. Prieto et al. (2025) demonstrated the dependence of grokking on regularization by showing that Softmax Collapse (i.e. floating point errors due to numerical instability) is responsible for the absence of grokking without regularization. We note that in addition to Prieto et al. (2025), several other papers considered grokking without explicit regularization (e.g., Xu et al. (2024); Levi et al. (2024)), but the grokking literature mostly focuses on settings with explicit regularization, as in this paper. Mallinar et al. (2025) revealed that grokking is not specific to neural networks nor to GD-based optimization methods by showing that grokking occurs when learning modular arithmetic with Recursive Feature Machines (RFM). Žunkovič & Ilievski (2024) analyzed grokking in linear classification, and showed quantitative bounds on the grokking time, but they did not provide rigorous proofs. Yunis et al. (2024) showed empirically that grokking is connected to rank minimization, namely, that the sudden drop in the test loss coincides precisely with the onset of low-rank behavior in the singular values of the weight matrices. Jeffares & van der Schaar (2025) argue that the settings in which grokking and other counterintuitive phenomena in deep learning can be shown to occur are limited in practice, as they may only appear in very specific situations.

## 1.2. Organization

The remaining of the paper is organized as follows. In Section 2, we formally describe the problem setup and introduce necessary notations. In Section 3, we provide an informal theorem to convey our main results in an accessible manner. In Section 4, we present our main results of end-to-end provable grokking in ridge regression. In Section 5, we support our theory with empirical verifications. In Section 6, we conclude the paper and propose some interesting future research directions. All proofs and additional experiments are deferred to the appendix.

## 2. Preliminaries

**Notations.** We use bold-face letters to denote vectors such as $\boldsymbol{x} = (x_1, \ldots, x_d) \in \mathbb{R}^d$ and use bold-face capital letters to denote matrices such as $\boldsymbol{X} = (\boldsymbol{x}_1, \ldots, \boldsymbol{x}_n)^\top \in \mathbb{R}^{n \times d}$.

We denote by $\boldsymbol{I}_d$ the $d$-dimensional identity matrix. For a vector $\boldsymbol{x}$, we denote by $\|\boldsymbol{x}\|_2$ its $\ell_2$ norm. For a matrix $\boldsymbol{X}$, we denote by $\|\boldsymbol{X}\|_{\text{op}}$ and $\|\boldsymbol{X}\|_F$ its operator and Frobenius norms, respectively. For a symmetric matrix $\boldsymbol{A}$, let $\lambda_{\min}(\boldsymbol{A})$ and $\lambda_{\min}^+(\boldsymbol{A})$ denote its smallest eigenvalue and its smallest positive eigenvalue, respectively. We use $\mathbb{1}\{\cdot\}$ to denote the indicator function, which equals one when its Boolean input is true and zero otherwise. We use standard asymptotic notations $O(\cdot)$ and $\Omega(\cdot)$ to hide numerical constant factors.

**Linear regression and data generation.** We consider a fundamental regression problem of learning a teacher function $N^*(\boldsymbol{x})$ where $\boldsymbol{x} \in \mathbb{R}^d$. The training data $\{(\boldsymbol{x}_i, y_i)\}_{i=1}^n \subseteq \mathbb{R}^d \times \mathbb{R}$ are generated according to the following: the input $\boldsymbol{x} \sim \mathcal{D}_{\boldsymbol{x}}$ follows some marginal distribution $\mathcal{D}_{\boldsymbol{x}}$ (which we do not specify), and is exactly labeled by the teacher function, i.e. $y = N^*(\boldsymbol{x})$. We train a student linear regression model

$$N(\boldsymbol{x}; \boldsymbol{\theta}) = \langle \boldsymbol{\theta}, \phi(\boldsymbol{x}) \rangle,$$

where $\phi(\boldsymbol{x}) : \mathbb{R}^d \mapsto \mathbb{R}^m$ is some fixed feature map and $\boldsymbol{\theta} \in \mathbb{R}^m$ is the trainable parameters, to learn the teacher. We assume that the teacher is realizable, i.e., $N^*(\boldsymbol{x}) = \langle \boldsymbol{\theta}^*, \phi(\boldsymbol{x}) \rangle$ for some $\boldsymbol{\theta}^* \in \mathbb{R}^m$.

**Ridge regression.** In a regression problem, it is natural to train the model by minimizing the empirical mean squared loss $\frac{1}{n} \sum_{i=1}^n (N(\boldsymbol{x}_i; \boldsymbol{\theta}) - N^*(\boldsymbol{x}_i))^2$ and evaluate the learning performance on the population squared loss $\mathbb{E}_{\boldsymbol{x}}[(N(\boldsymbol{x}; \boldsymbol{\theta}) - N^*(\boldsymbol{x}))^2]$. We use ridge regression by appending an $\ell_2$-regularization term to the loss function. Let $L_n(\boldsymbol{\theta}; \lambda)$ denote the regularized mean squared training error, let $L_n(\boldsymbol{\theta})$ denote the unregularized mean squared loss, and let $L_\lambda(\boldsymbol{\theta})$ denote the $\ell_2$ penalty. Our training objective is

$$L_n(\boldsymbol{\theta}; \lambda) = L_n(\boldsymbol{\theta}) + L_\lambda(\boldsymbol{\theta})$$
$$= \frac{1}{2n} \sum_{i=1}^n (N(\boldsymbol{x}_i; \boldsymbol{\theta}) - N^*(\boldsymbol{x}_i))^2 + \frac{\lambda}{2} \|\boldsymbol{\theta}\|_2^2, \quad (1)$$

where $\lambda > 0$ is the regularization parameter. We scale the loss objective by a constant factor of $1/2$ for the simplicity of gradient analysis. We optimize Equation (1) using the vanilla Gradient Descent (GD) algorithm, which updates $\boldsymbol{\theta}$ via

$$\boldsymbol{\theta}^{(t+1)} = \boldsymbol{\theta}^{(t)} - \eta \nabla_{\boldsymbol{\theta}} L_n(\boldsymbol{\theta}^{(t)}; \lambda), \quad \forall t \in \mathbb{N},$$

where $\eta > 0$ is the step-size or learning rate. While we focus on GD with a fixed step size, we note that our analysis should also hold for GD with a decaying step size and for gradient flow.

When $N(\boldsymbol{x}; \boldsymbol{\theta}) = \langle \boldsymbol{\theta}, \phi(\boldsymbol{x}) \rangle$ and $N^*(\boldsymbol{x}) = \langle \boldsymbol{\theta}^*, \phi(\boldsymbol{x}) \rangle$, our loss objective in Equation (1) can be simplified to

$$L_n(\boldsymbol{\theta}; \lambda) = \frac{1}{2n} \sum_{i=1}^n (\langle \boldsymbol{\theta} - \boldsymbol{\theta}^*, \phi(\boldsymbol{x}) \rangle)^2 + \frac{\lambda}{2} \|\boldsymbol{\theta}\|_2^2,$$

and each GD iteration realizes an update in the form of

$$\boldsymbol{\theta}^{(t+1)} = \boldsymbol{\theta}^{(t)} - \eta \nabla_{\boldsymbol{\theta}} L_n(\boldsymbol{\theta}^{(t)}; \lambda)$$
$$= \boldsymbol{\theta}^{(t)} - \frac{\eta}{n} \sum_{i=1}^n \left\langle \boldsymbol{\theta}^{(t)} - \boldsymbol{\theta}^*, \phi(\boldsymbol{x}_i) \right\rangle \phi(\boldsymbol{x}_i) - \eta \lambda \boldsymbol{\theta}^{(t)} \quad (2)$$

To present our results, we introduce the following additional notations. Let $L(\boldsymbol{\theta}) = \mathbb{E}_{\boldsymbol{x}}[(N(\boldsymbol{x}; \boldsymbol{\theta}) - N^*(\boldsymbol{x}))^2]$ denote the generalization (population) squared loss. We define the empirical feature map as $\boldsymbol{\Phi} = (\phi(\boldsymbol{x}_1), \ldots, \phi(\boldsymbol{x}_n))^\top \in \mathbb{R}^{n \times m}$. Let $L = \frac{1}{n} \|\boldsymbol{\Phi}\|_F^2 = \frac{1}{n} \sum_{i=1}^n \|\phi(\boldsymbol{x}_i)\|_2^2$ denote the (average) norm of the feature vectors. Let $\boldsymbol{\Sigma} = \mathbb{E}_{\boldsymbol{x}}[\phi(\boldsymbol{x})\phi(\boldsymbol{x})^\top] \in \mathbb{R}^{m \times m}$ denote the feature population covariance matrix.

## 3. Informal Results

To demonstrate a clear manifestation of grokking, our main theorem is formulated as follows: GD achieves a training error smaller than $\epsilon > 0$ in an early stage of training (at step $t_1$), while a poor generalization error well above some constant $c > 0$ persists far beyond the point of overfitting (until step $t_2 > t_1$), yet eventually a generalization error smaller than $\epsilon$ is reached. In particular, this establishes grokking when $c \geq \epsilon$. Our results give a lower bound on $(t_2 - t_1)$ in terms of both $\epsilon$ and $c$, where this difference can be made arbitrarily large via proper hyperparameter tuning for any choice of $\epsilon$ and $c \geq \epsilon$. Specifically, let $t_1$ be the largest number of training steps such that the empirical squared loss is above $\epsilon$, and let $t_2$ be the smallest number of training steps for which the generalization loss is below $c$. Our main results are summarized by the following informal theorem, establishing end-to-end provable grokking in ridge regression.

**Informal Theorem** (**End-to-end provable grokking for ridge regression**). *Consider a linear regression problem of training a student model $N(\boldsymbol{x}; \boldsymbol{\theta}) = \langle \boldsymbol{\theta}, \phi(\boldsymbol{x}) \rangle$ to learn a realizable teacher $N^*(\boldsymbol{x}) = \langle \boldsymbol{\theta}^*, \phi(\boldsymbol{x}) \rangle$, where $\boldsymbol{\theta} \in \mathbb{R}^m$ is the trainable parameter and $\boldsymbol{\theta}^*$ is unknown. Suppose that the training is done by optimizing an MSE over $n$ samples using randomly initialized gradient descent at $\boldsymbol{\theta}^{(0)}$ with weight decay parameter $\lambda$.*

*Under the initialization $\boldsymbol{\theta}^{(0)} \sim \mathcal{N}(\boldsymbol{0}, \nu^2 \boldsymbol{I}_m)$ for some $\nu^2 > 0$, and some mild distributional assumptions on the feature map $\phi(\boldsymbol{x})$, we have with high probability that for a sufficiently large sample size $n$, and a sufficiently large feature space dimensionality $m$, $(t_2 - t_1)$ can be lower bounded by an arbitrarily large value by choosing a sufficiently small weight decay $\lambda$.*

## 4. Grokking in Ridge Regression

We now formalize our end-to-end provable grokking results for ridge regression problems. For any realizable teacher

function, we provide a quantitative lower bound for the grokking time $(t_2 - t_1)$ in terms of training hyperparameters, as well as bounds on the hyperparameters that suffice to realize grokking. Based on these bounds, we analyze how the different hyperparameters affect grokking, which is corroborated in our experiments in Section 5.

### 4.1. Warmup: Grokking with the Zero Teacher

We start by considering a simple scenario where the target teacher is the zero function. When $N^*(\boldsymbol{x}) = 0$, we have

$$L_n(\boldsymbol{\theta}; \lambda) = \frac{1}{2n} \sum_{i=1}^n (N(\boldsymbol{x}_i; \boldsymbol{\theta}))^2 + \frac{\lambda}{2} \|\boldsymbol{\theta}\|_2^2;$$

$$\boldsymbol{\theta}^{(t+1)} = \boldsymbol{\theta}^{(t)} - \frac{\eta}{n} \sum_{i=1}^n \langle \boldsymbol{\theta}^{(t)}, \phi(\boldsymbol{x}_i) \rangle \phi(\boldsymbol{x}_i) - \eta \lambda \boldsymbol{\theta}^{(t)}.$$

The following theorem describes an end-to-end provable grokking result for learning the zero teacher in a ridge regression setting using gradient descent. To develop some initial intuition on what causes grokking, we state the aforementioned three stages of grokking in terms of the convergence rates of the training/generalization squared losses, in contrast to the format of our Theorem 4.2 which appears later.

**Theorem 4.1** (**End-to-end provable grokking for the zero teacher**). *Suppose that the teacher function is $N^*(\boldsymbol{x}) = 0$. Consider training a student $N(\boldsymbol{x}; \boldsymbol{\theta}) = \langle \boldsymbol{\theta}, \phi(\boldsymbol{x}) \rangle$ to learn the teacher via optimizing the ridge regression objective in Equation (1) using randomly initialized GD with $\boldsymbol{\theta}^{(0)} \sim \mathcal{N}(\boldsymbol{0}, \nu^2 \boldsymbol{I}_m)$. Assume that $\eta \leq 2/(L + 2\lambda)$. Then for any $t \in \mathbb{N}$, we have*

(i) $L_n(\boldsymbol{\theta}^{(t)}) \leq \frac{L}{2} \cdot (1 - \frac{1}{n}\eta\lambda_{\min}^+(\boldsymbol{\Phi}^\top\boldsymbol{\Phi}) - \eta\lambda)^{2t} \cdot \|\boldsymbol{\theta}^{(0)}\|_2^2$;
(ii) *With probability at least $1 - 2e^{-(m-n)/32}$,*
    $L(\boldsymbol{\theta}^{(t)}) \geq \lambda_{\min}(\boldsymbol{\Sigma}) \cdot (1 - \eta\lambda)^{2t} \cdot \frac{(m-n)\nu^2}{2}$;
(iii) $\|\boldsymbol{\theta}^{(t)}\|_2^2 \leq (1 - \eta\lambda)^{2t} \cdot \|\boldsymbol{\theta}^{(0)}\|_2^2$.

Studying the zero teacher as a warmup yields the following advantages. First, choosing the zero teacher enables us to build a much simpler generalization guarantee. Clearly, item (iii) above implies that GD eventually converges to the solution $\boldsymbol{\theta}^* = \boldsymbol{0}$. Since $N^*(\boldsymbol{x}) = 0$, this yields generalization. Additionally, choosing the zero teacher allows us to derive matching lower (ii) and upper (iii) bounds for the generalization loss in terms of the convergence rate.

More importantly, Theorem 4.1 reveals that the training and test losses can decrease at different rates, and the potentially considerable discrepancy between these rates causes grokking. Specifically, given the empirical feature map $\boldsymbol{\Phi}^\top\boldsymbol{\Phi}$, a sufficiently small regularization $\lambda$ can only have a negligible effect on the convergence rate of the training MSE. This is because the convergence rate

of $(1 - \frac{1}{n}\eta\lambda_{\min}^+(\boldsymbol{\Phi}^\top\boldsymbol{\Phi}) - \eta\lambda)^{2t}$ will be dominated by the term $(1 - \frac{1}{n}\eta\lambda_{\min}^+(\boldsymbol{\Phi}^\top\boldsymbol{\Phi}))^{2t}$ when $\lambda$ is sufficiently small. However, decreasing $\lambda$ can make the generalization convergence rate $(1 - \eta\lambda)^{2t}$ arbitrarily slow. In conclusion, Theorem 4.1 establishes a provable instance of grokking whenever $m \gg n$, $\lambda \ll \frac{1}{n}\lambda_{\min}^+(\boldsymbol{\Phi}^\top\boldsymbol{\Phi})$ and $\lambda \to 0$.

### 4.2. Grokking with Realizable Teachers

For the most general setting studied in this paper, we consider the realizable case where the ground-truth teacher is $N^*(\boldsymbol{x}) = \langle \boldsymbol{\theta}^*, \phi(\boldsymbol{x}) \rangle$ for some unknown $\boldsymbol{\theta}^* \in \mathbb{R}^m$, and the student is $N(\boldsymbol{x}; \boldsymbol{\theta}) = \langle \boldsymbol{\theta}, \phi(\boldsymbol{x}) \rangle$ where $\boldsymbol{\theta} \in \mathbb{R}^m$ is the vector of trainable parameters. We show that for any teacher, there is a realization of training hyperparameters that results in an arbitrarily long grokking time. Our main result is the following.

**Theorem 4.2** (**End-to-end provable grokking for realizable ridge regression**). *Assume a realizable teacher function $N^*(\boldsymbol{x}) = \langle \boldsymbol{\theta}^*, \phi(\boldsymbol{x}) \rangle$ for some $\boldsymbol{\theta}^* \in \mathbb{R}^m$. Assume the boundedness of the feature map, i.e. $\|\phi(\boldsymbol{x})\|_2 \leq b$ for any $\boldsymbol{x}$ and some $b > 0$. Consider training a student $N(\boldsymbol{x}; \boldsymbol{\theta}) = \langle \boldsymbol{\theta}, \phi(\boldsymbol{x}) \rangle$ to learn the teacher via optimizing the ridge regression objective in Equation (1) using randomly initialized GD with $\boldsymbol{\theta}^{(0)} \sim \mathcal{N}(\boldsymbol{0}, \nu^2 \boldsymbol{I}_m)$. For all $\epsilon > 0$, let $c \geq \epsilon$ be arbitrary. We define*

$$t_1 := t_1(\epsilon) := \max\left(\left\{t \in \mathbb{N} : L_n(\boldsymbol{\theta}^{(t)}) \geq \epsilon\right\}\right)$$

*and*

$$t_2 := t_2(c) := \min\left(\left\{t \in \mathbb{N} : L(\boldsymbol{\theta}^{(t)}) \leq c\right\}\right).$$

*For any $\delta \in (0, 1)$, assume a sufficiently large sample size:*

$$n = \Omega\left(\frac{b^4 \|\boldsymbol{\theta}^*\|_2^4}{\epsilon^2} \log\left(\frac{1}{\delta}\right)\right), \tag{3}$$

*a sufficiently large dimensionality of the feature map:*

$$m = n + \Omega\left(\max\left\{\log\left(\frac{1}{\delta}\right), \frac{\|\boldsymbol{\theta}^*\|_2^2}{\nu^2}, \frac{c}{\lambda_{\min}(\boldsymbol{\Sigma})\nu^2}\right\}\right), \tag{4}$$

*and finally, a sufficiently small weight decay:*

$$\lambda = O\left(\min\left\{\frac{\epsilon}{\|\boldsymbol{\theta}^*\|_2^2}, \frac{b\sqrt{\epsilon}}{\|\boldsymbol{\theta}^*\|_2}\right\}\right). \tag{5}$$

*Then, if $\eta < 1/(\lambda + b^2)$, we have with probability at least $1 - \delta$ that*

$$t_1 \leq \frac{n \ln\left(\frac{6b^2\|\boldsymbol{\theta}^{(0)}\|_2^2}{\epsilon}\right)}{2\eta\lambda_{\min}^+(\boldsymbol{\Phi}^\top\boldsymbol{\Phi})} \tag{6}$$

*and*

$$t_2 \geq \frac{\ln\left(\frac{(m-n)\nu^2}{2}\left(\sqrt{\frac{c}{\lambda_{\min}(\boldsymbol{\Sigma})}} + \|\boldsymbol{\theta}^*\|_2\right)^{-2}\right)}{4\eta\lambda}. \tag{7}$$

*Moreover, $L(\boldsymbol{\theta}^{(t)}) \le \epsilon$ for all sufficiently large $t$.*

The main intuition behind this discrepancy between $t_1$ and $t_2$ can be explained as follows: When the student regression model is sufficiently over-parameterized ($m \gg n$), we are solving a high-dimensional regression problem. Therefore, when the weight decay $\lambda$ is small, the GD optimization process only effectively updates the projection of the weight vector $\boldsymbol{\theta}_{\parallel}$ onto the data-spanning subspace to fit the training data, while having negligible effect on the component $\boldsymbol{\theta}_{\perp}$ in the complementary subspace, which remains close to its initialization. This occurs since the information captured by the data only contributes to the convergence of $\boldsymbol{\theta}_{\parallel}$ at a rate of $(1 - \frac{1}{n}\eta\lambda_{\min}^{+}(\boldsymbol{\Phi}^{\top}\boldsymbol{\Phi}) - \eta\lambda)^t$. In contrast, $\boldsymbol{\theta}_{\perp}$ converges at a negligible rate of $(1 - \eta\lambda)^t$ as a consequence of long-term weight decay. Such an imbalanced trajectory prevents timely generalization and leads to harmful overfitting. As a remedy, weight decay plays a crucial role in decreasing the complexity of the model class, which eventually guarantees uniform convergence and thus good generalization.

Next, we disentangle the influence of individual hyperparameters on the grokking time according to our theory, which aligns closely with our experimental results in Section 5. Notably, these functional dependencies on the hyperparameters are not only qualitatively corroborated by our experiments, but they also quantitatively predict the expected scaling rates (see Figure 2).

Evidently, Equations (6) and (7) together provide strong control of the grokking time $(t_2 - t_1)$, which reveals the following effects of hyperparameter tuning on grokking:

- **Weight decay** $\lambda$: For small $\lambda$, decreasing $\lambda$ increases $t_2$ with $t_2 \propto 1/\lambda$ when other hyperparameters are held fixed, and $\lambda$ has no effect on our bound for $t_1$. Therefore, $(t_2 - t_1) \to \infty$ as $\lambda \to 0$. We remark that for large $\lambda$ (which is not our focus in this work), our upper bound in Equation (6) is not tight in general, as a tight bound should incorporate the term $\eta\lambda_{\min}^{+}(\boldsymbol{\Phi}^{\top}\boldsymbol{\Phi}) + \eta\lambda$ in the denominator.

- **Sample size** $n$ **and feature dimensionality** $m$: It is difficult to deduce clean dependencies that capture the behavior of $t_1$ as a function of $n$ and $m$, since there is no known quantitative bound revealing how $\lambda_{\min}^{+}(\boldsymbol{\Phi}^{\top}\boldsymbol{\Phi})$ depends on $m$ and $n$, even when making strong assumptions on the distribution of the features (see Remark 4.3 for a discussion). On a more positive note, we show empirically that the behavior of $t_1$ as a function of $n$ and $m$ strongly accords with our theoretical bound in Equation (6).

  As for $t_2$, since we make no assumption on the feature map $\phi(\boldsymbol{x})$, the quantity $\lambda_{\min}(\boldsymbol{\Sigma})$ is not necessarily controlled. Indeed, more explicit dependencies of $t_2$ on $n$ and $m$ can be derived by assuming distributional assumptions. For instance, if $\phi(\boldsymbol{x}_1), \ldots, \phi(\boldsymbol{x}_n)$

are i.i.d. spherical Gaussian vectors. We present such clean bounds in Equation (8) and show that it strongly matches our experimental results.

- **Initialization scale** $\nu^2$: Increasing $\nu^2$ increases $t_1$ and $t_2$ simultaneously with $t_1, t_2 \propto \ln(\nu^2)$. More precisely, when $\lambda$ is sufficiently small, we have $t_2 > t_1$, and moreover $t_1 \le c_1 \ln(\nu^2) + a_1$ and $t_2 \ge c_2 \ln(\nu^2) + a_2$ for some $c_2 > c_1 > 0$ and some $a_1, a_2$. In such a scenario, increasing $\nu^2$ increases $(t_2 - t_1)$ and amplifies grokking.

**Remark 4.3** (on $\lambda_{\min}^{+}(\boldsymbol{\Phi}^{\top}\boldsymbol{\Phi})$ **and its dependence on** $n$ **and** $m$). $\lambda_{\min}^{+}(\boldsymbol{\Phi}^{\top}\boldsymbol{\Phi})$ *is an empirical quantity which depends on the sampled features. We note that even for simple distributions such as $\phi(\boldsymbol{x}) \sim \mathcal{N}(\boldsymbol{0}, \frac{1}{m}\boldsymbol{I}_m)$, the behavior of $\lambda_{\min}^{+}(\boldsymbol{\Phi}^{\top}\boldsymbol{\Phi})$ as a function of $n$ and $m$ is not yet fully understood. While there is a known asymptotic result given by the celebrated Marchenko–Pastur law (Marčenko & Pastur, 1967), showing that when $m, n \to \infty$ and $m/n \to \gamma$ for some $\gamma \in (0, \infty)$, $\lambda_{\min}^{+}(\boldsymbol{\Phi}^{\top}\boldsymbol{\Phi}) \to \gamma^{-1}(1 - \sqrt{\gamma})^2$, to the best of our knowledge, there is no quantitative bound revealing how this depends on $m$ and $n$.*

Next, we state a separate theorem for each of the three stages of our grokking result, as it is helpful for understanding each hyperparameter effect separately.

**Theorem 4.4** (**Training loss convergence**). *Suppose that $N^*(\boldsymbol{x}) = \langle \boldsymbol{\theta}^*, \phi(\boldsymbol{x}) \rangle$ for some $\boldsymbol{\theta}^* \in \mathbb{R}^m$, and that $\boldsymbol{\theta}^{(0)} \sim \mathcal{N}(\boldsymbol{0}, \nu^2 \boldsymbol{I}_m)$. Assume that $\eta < 1/(\lambda + \lambda_{\max}(\boldsymbol{\Phi}^{\top}\boldsymbol{\Phi})/n)$. For any $\epsilon > 0$ and $\delta \in (0, 1)$, suppose that*

$$m = \Omega\left(\max\left\{\log\left(\frac{1}{\delta}\right), \frac{\|\boldsymbol{\theta}^*\|_2^2}{\nu^2}\right\}\right)$$

*and*

$$\lambda = O\left(\min\left\{\frac{\epsilon}{\|\boldsymbol{\theta}^*\|_2^2}, \frac{\sqrt{L\epsilon}}{\|\boldsymbol{\theta}^*\|_2}\right\}\right).$$

*Then, with probability at least $1 - \delta$, we have $L_n(\boldsymbol{\theta}^{(t)}) \le \epsilon$ for all*

$$t \ge \frac{1}{2}\log_{1 - \frac{\eta\lambda_{\min}^{+}(\boldsymbol{\Phi}^{\top}\boldsymbol{\Phi})}{n} - \eta\lambda}\left(\frac{\epsilon}{6L\|\boldsymbol{\theta}^{(0)}\|_2^2}\right).$$

The above assumption that $m = \Omega(\|\boldsymbol{\theta}^*\|_2^2/\nu^2)$ implies that the student is initialized to have a large norm with respect to the teacher's norm. Since we aim to upper bound $L_n(\boldsymbol{\theta}^{(t)})$ and not $L_n(\boldsymbol{\theta}^{(t)}; \lambda)$, the assumed upper bound on $\lambda$ is for technical purposes, and it does not limit our results since small values of $\lambda$ facilitate grokking.

**Theorem 4.5** (**Poor generalization when overfitting**). *Suppose that $N^*(\boldsymbol{x}) = \langle \boldsymbol{\theta}^*, \phi(\boldsymbol{x}) \rangle$ for some $\boldsymbol{\theta}^* \in \mathbb{R}^m$, that $\boldsymbol{\theta}^{(0)} \sim \mathcal{N}(\boldsymbol{0}, \nu^2 \boldsymbol{I}_m)$, and further assume that $\eta < 1/\lambda$. For any $c > 0$ and $\delta \in (0, 1)$, suppose that*

$$m = n + \Omega\left(\max\left\{\log\left(\frac{1}{\delta}\right), \frac{\|\boldsymbol{\theta}^*\|_2^2}{\nu^2}, \frac{c}{\lambda_{\min}(\boldsymbol{\Sigma})\nu^2}\right\}\right).$$

*Then, with probability at least $1 - \delta$, we have $L(\boldsymbol{\theta}^{(t)}) \geq c$ for all*

$$t \leq \log_{1-\eta\lambda}\left(\sqrt{\frac{2}{(m-n)\nu^2}}\left(\sqrt{\frac{c}{\lambda_{\min}(\boldsymbol{\Sigma})}} + \|\boldsymbol{\theta}^*\|_2\right)\right).$$

The lower bound on $m$ here is stronger than the one in Theorem 4.4, which ensures that we are in a high-dimensional regression situation. In our last theorem, we prove that good generalization eventually occurs.

**Theorem 4.6 (Generalization).** *Suppose that $N^*(\boldsymbol{x}) = \langle\boldsymbol{\theta}^*, \boldsymbol{\phi}(\boldsymbol{x})\rangle$ for some $\boldsymbol{\theta}^* \in \mathbb{R}^m$. Assume that $\|\boldsymbol{\phi}(\boldsymbol{x})\|_2 \leq b$ for any $\boldsymbol{x}$ for some $b > 0$. Let $\boldsymbol{\theta}_\lambda^* = \arg\min_{\boldsymbol{\theta}}\{L_n(\boldsymbol{\theta}; \lambda)\}$. For any $\epsilon > 0$ and $\delta \in (0, 1)$, suppose that*

$$n = \Omega\left(\frac{b^4\|\boldsymbol{\theta}^*\|_2^4}{\epsilon^2}\log\left(\frac{1}{\delta}\right)\right).$$

*Then, if $\eta \leq 1/(\lambda + b^2)$, GD converges to $\boldsymbol{\theta}_\lambda^*$ satisfying $L(\boldsymbol{\theta}_\lambda^*) \leq 2L_n(\boldsymbol{\theta}_\lambda^*) + \epsilon$, with probability at least $1 - \delta$.*

The lower bound on the sample size $n = \Omega(b^4\|\boldsymbol{\theta}^*\|_2^4\epsilon^{-2})$ stems from a standard uniform convergence argument based on Rademacher complexity.

# 5. Experiments

In this section, we systematically verify our theoretical understanding of grokking via experiments. Throughout, we use $n, \eta$ and $\lambda$ to denote the training sample size, GD step size and weight decay parameter, respectively. In all of our experiments, we set the error thresholds $c = \epsilon = 0.01$.

## 5.1. Hyperparameter Control

We empirically verify our quantitative bounds on how the model's hyperparameters affect grokking in realizable ridge regression. Specifically, we train a student linear regression model using gradient descent with weight decay, to learn a realizable teacher with unit norm, i.e. $\|\boldsymbol{\theta}^*\|_2 = 1$. We simply choose the feature map to be the identity function, i.e. $\boldsymbol{\phi}(\boldsymbol{x}) = \boldsymbol{x} \in \mathbb{R}^m$. We use $\nu^2$ to denote the initialization scale of weights in line with our theory.

As discussed in Section 4, $\lambda_{\min}(\boldsymbol{\Sigma})$ cannot be controlled without making additional assumptions on $\boldsymbol{\phi}(\boldsymbol{x})$. Consequentially, by introducing additional distributional assumptions, we are able to provide more explicit bounds for $t_1$ and $t_2$ based on Theorem 4.2. Specifically, we assume that $\boldsymbol{\phi}(\boldsymbol{x}_1), \ldots, \boldsymbol{\phi}(\boldsymbol{x}_n)$ are drawn i.i.d. from a Gaussian distribution $\mathcal{N}(\boldsymbol{0}, \frac{1}{m}\boldsymbol{I}_m)$. In such a case, if $\eta\lambda \leq 0.01$, a simple calculation (see Remark A.14 for details) yields that

$$t_2 \geq \frac{\ln\left(\frac{(m-n)\nu^2}{8m\epsilon}\right)}{2.02\eta\lambda} \quad \text{and} \quad t_1 \leq \frac{n\ln\left(\frac{14m\nu^2}{\epsilon}\right)}{2\eta\lambda_{\min}^+(\boldsymbol{\Phi}^\top\boldsymbol{\Phi})}. \quad (8)$$

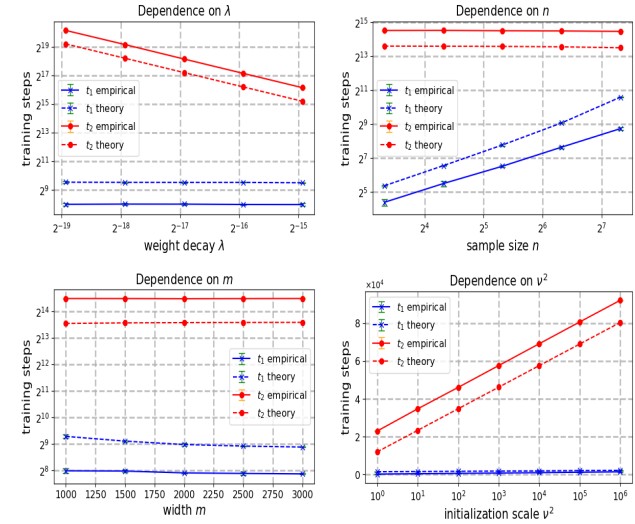

*Figure 2.* Plots for the effects of hyperparameters on grokking in ridge regression. **Left Upper:** decreasing weight decay extends the generalization delay ($t_2 \propto 1/\lambda$); **Right Upper:** decreasing sample size amplifies grokking by speeding up the convergence of the training loss (affects $t_1$); **Left Lower:** increasing the feature dimension has little effect on $t_1$ and $t_2$; **Right Lower:** increasing the initialization scale increases $t_1$ and $t_2$ simultaneously at logarithmic rates ($t_1, t_2 \propto \ln(\nu^2)$).

Our experiments are also implemented with such a distributional assumption. We set $\eta = 1$, and unless otherwise stated for comparison purposes, we use the default values $n = 100$, $m = 1000$, $\nu^2 = 1$ and $\lambda = 10^{-4}$.

The results are shown in Figure 2, where the dashed lines represent our theoretical bounds for $t_1$ and $t_2$ in Equation (8), and solid lines represent the values of $t_1$ and $t_2$ that were observed empirically. Notably, all pairs of solid and dashed lines seem to match in their behavior, implying that our theoretical bounds provide strong control.

## 5.2. Random-Features Neural Networks

In this subsection, we train a two-layer random ReLU features network to learn a single ReLU neuron teacher network with unit norm weights, i.e. $\boldsymbol{x} \mapsto \sigma(\langle\boldsymbol{w}^*, \boldsymbol{x}\rangle)$ with $\|\boldsymbol{w}^*\|_2 = 1$, where $\sigma(\cdot) = \max\{0, \cdot\}$ is the ReLU activation. A two-layer ReLU neural network is in the form of $N(\boldsymbol{x}; \boldsymbol{\theta}) = \sum_{j=1}^m a_j\sigma(\langle\boldsymbol{w}_j, \boldsymbol{x}\rangle)$ for all $\boldsymbol{x} \in \mathbb{R}^d$. For random-features neural networks, we only optimize the output layer weights $\boldsymbol{\theta} = \boldsymbol{a}$, and the hidden layer weights are initialized and then fixed during training. Clearly, this random feature model is a special case of linear regression with feature map $\boldsymbol{\phi}(\boldsymbol{x}) = (\boldsymbol{\phi}_1(\boldsymbol{x}), \ldots, \boldsymbol{\phi}_m(\boldsymbol{x}))$ where $\boldsymbol{\phi}_j(\boldsymbol{x}) = \sigma(\langle\boldsymbol{w}_j, \boldsymbol{x}\rangle)$ for all $j \in [m]$. We use the following initialization scheme $a_j^{(0)} \sim \mathcal{N}(0, 1), \boldsymbol{w}_j \sim \mathcal{N}(\boldsymbol{0}, \frac{\nu^2}{dm}\boldsymbol{I}_d)$ for each $j \in [m]$.

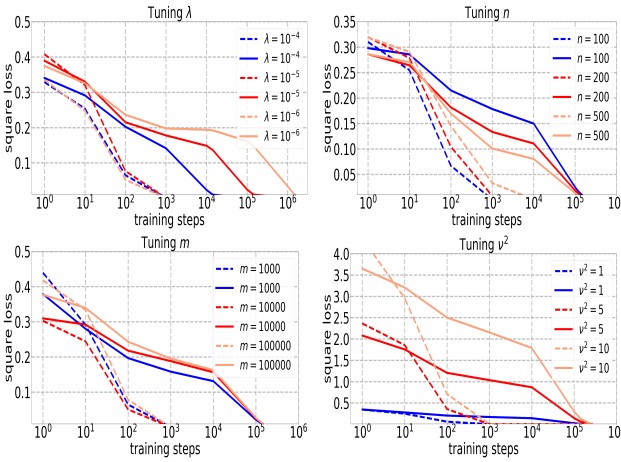

*Figure 3.* Plots of training and test losses of a two-layer random ReLU features network. Dashed/solid lines indicate train/test loss respectively. **Left Upper:** using smaller weight decay amplifies the grokking time by delaying generalization (increases $t_2$); **Right Upper:** having smaller sample size amplifies grokking by speeding up the training convergence (decreases $t_1$); **Left Lower:** increasing the student's width does not significantly prolong grokking; **Left Lower:** increasing the initialization scale does not significantly prolong grokking, but instead widens the gap between the generalization and training losses during the overfitting stage.

We present the results in Figure 3, by a standard comparison between the training and test losses. We set: $d = 100$, $\eta = 1$, and unless otherwise stated for comparison purposes, we use the default values $n = 100$, $\lambda = 10^{-5}$, $m = 10000$ and $\nu^2 = 1$. Since the teacher $\boldsymbol{w}^*$ is sampled independently of the student network, this sets us in a non-realizable setting. However, since a sufficiently wide student network can approximate any teacher to arbitrary accuracy with high probability, we are essentially arbitrarily close to the realizable setting with sufficient over-parameterization. The figure shows behavior similar to that of realizable ridge regression.

### 5.3. Non-linear Neural Networks

We also study empirically the effects of the different hyperparameters on grokking for training (both layers of) a two-layer ReLU neural network as defined in Subsection 5.2 with $\boldsymbol{\theta} = (\boldsymbol{W}, \boldsymbol{a})$. We use the following initialization scheme $a_j^{(0)} \sim \mathcal{N}(0, \frac{1}{m}), \boldsymbol{w}_j^{(0)} \sim \mathcal{N}(\boldsymbol{0}, \frac{\nu^2}{d}\boldsymbol{I}_d)$ for each $j \in [m]$. Specifically, we choose the zero function to be the underlying teacher, and set $\eta = 10^{-4}, d = 50$, and unless otherwise stated for comparison purposes, we use the default values $n = 50$, $m = 1000$, $\nu^2 = 1$ and $\lambda = 0.05$.

The results are presented in Figure 4. Notably, the dependencies of $t_1$ and $t_2$ on the hyperparameters match qualitatively to those in Figure 2. This suggests that our bounds might capture behaviors that hold much more generally than our

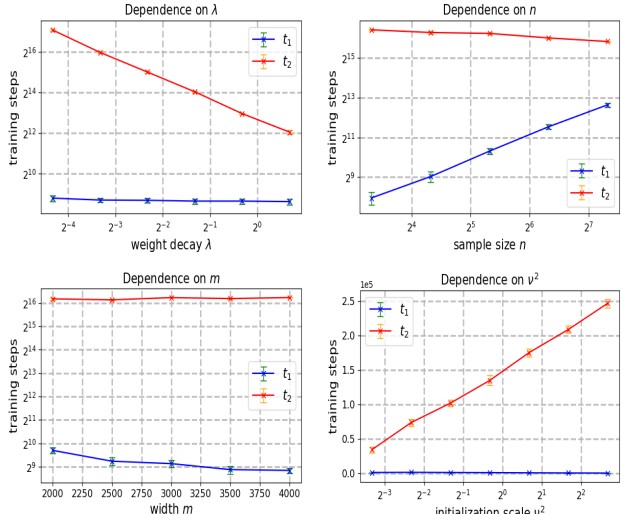

*Figure 4.* Plots for the effect of the hyperparameters on grokking when training a two-layer ReLU network with the zero teacher.

currently analyzed setting, and thus exploring it further is an intriguing direction for future work.

Finally, we include the experimental setup for Figure 1 for completeness. **Left**: $n = 100$, $m = 10000$, $\lambda = 10^{-5}$, $\nu^2 = 1$ and $\eta = 1$; **Right**: $d = 50$, $n = 50$, $m = 1000$, $\lambda = 0.1$, $\nu^2 = 1$ and $\eta = 10^{-4}$. See Appendix B for additional experiments.

## 6. Discussion and Future Directions

We study grokking and establish the first end-to-end provable grokking result for the classical ridge regression model. We derive quantitative bounds for the grokking time in terms of the model hyperparameters and verify our theoretical findings via experiments. Our work establishes a rigorous theoretical foundation for grokking. Observing this phenomenon within the fundamental framework of linear regression is not only surprising in its simplicity, but also highly illuminating; it provides a controlled environment that allows us to cleanly isolate the precise effects of individual hyperparameters. By uncovering the root causes of grokking in this transparent setting, our analysis may serve as a stepping stone toward demystifying its dynamics in modern deep learning architectures.

Our work leaves several interesting questions open. Firstly, can we show end-to-end provable grokking results for non-realizable ridge regression? This includes the special case of ridge regression with label noise (see Figure 6 in Appendix B). Since we have already observed grokking empirically in non-realizable learning with two-layer random-features neural networks, it is natural to start studying such random features models. Indeed, deriving provable

grokking results for learning agnostically with two-layer ReLU random features neural networks seems challenging. We find that current theoretical analyses fail to identify a hyperparameter realization for grokking. Specifically, when using our techniques, choosing an arbitrarily small weight decay parameter in the agnostic learning setting may impede generalization entirely. This suggests that new tools and more careful convergence analyses might be required.

Another question left open by our research is whether we can show end-to-end provable grokking as a result of a transition from the lazy to the rich regime of training neural networks. As discussed earlier, many existing works attribute grokking to such lazy-to-rich regime transitions, and have verified it empirically, but none provided a rigorous theoretical analysis proving it. Motivated by this, we have also explored grokking in a setting where we learn a target neural network teacher, by training a two-layer student network using GD with weight decay. Unlike the ridge regression setting addressed in this paper, proving that overfitting occurs remains challenging even under strong assumptions, such as a zero-teacher model where GD only optimizes the hidden layer weights. While a classical neural tangent kernel (NTK) analysis implies that each neuron remains close to its initialization during training, this does not guarantee persistent poor generalization; the collective contribution of $m$ neurons may still allow the trained network to converge toward the zero function.

We note that when training with a small weight decay coefficient, the overfitting solution that we obtain corresponds to training without weight decay, and the asymptotic solution is the minimum-norm interpolating predictor, which corresponds to training with zero initialization without weight decay (Gunasekar et al., 2018; Vardi & Shamir, 2021). Hence, in our setting grokking corresponds to a transition from a misspecified to a well-specified prior. Moreover, Lauditi et al. (2025) showed that weight decay in wide non-linear neural networks may cause the initial NTK to decay leaving only a data-dependent NTK. Thus, similarly to our work, in their setting weight decay changes an implicit prior. However, it is unclear whether their prior change can lead to grokking.

Finally, it is worth noting that grokking in regression tasks exhibits structural differences from its classification counterpart, particularly regarding the characteristic "test error plateau". In classification, this plateau is typically an artifact of evaluating a discrete metric (accuracy) rather than a continuous loss (e.g. Gromov, 2023, Figure 1). In regression, however, the continuous loss is the natural primary metric, which typically decays smoothly without an explicit plateau. To bridge this gap and investigate whether our setting captures this classic phenomenon, we define a threshold-based surrogate accuracy metric given

by $\mathbb{P}_{\boldsymbol{x}}((N(\boldsymbol{x}; \boldsymbol{\theta}^{(t)}) - N^*(\boldsymbol{x}))^2 \leq \epsilon)$ for a fixed $\epsilon > 0$. Remarkably, evaluating our model under this metric reveals the familiar plateauing phenomenon experimentally (see Figure 5 in Appendix B). This confirms that the plateau is not unique to classification settings but can be recovered in linear regression under appropriate evaluation, offering a promising direction for future theoretical analysis.

## Acknowledgements

We thank the anonymous reviewers who provided useful suggestions to improve the quality of this paper. GV is supported by the Israel Science Foundation (grant No. 2574/25), by a research grant from Mortimer Zuckerman (the Zuckerman STEM Leadership Program), and by research grants from the Center for New Scientists at the Weizmann Institute of Science, and the Shimon and Golde Picker – Weizmann Annual Grant. IS is supported by the Israel Science Foundation (grant No. 1753/25).

## Impact Statement

This paper presents work whose goal is to advance the field of Machine Learning. Since this work is mainly theoretical in its nature, there are no societal implications that require discloser as far as we can discern.

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

# A. Missing Proofs

**Theorem A.1** (**Theorem 4.1 restated**). *Assume that $0 < \eta \leq 2/(L + 2\lambda)$ and $\boldsymbol{\theta}^{(0)} \sim \mathcal{N}(\mathbf{0}, \nu^2 \boldsymbol{I}_m)$. Then for any $t \in \mathbb{N}$, we have*

1. $L_n(\boldsymbol{\theta}^{(t)}) \leq \frac{L}{2} \cdot (1 - \frac{1}{n}\eta\lambda_{\min}^+(\boldsymbol{\Phi}^\top\boldsymbol{\Phi}) - \eta\lambda)^{2t} \cdot \|\boldsymbol{\theta}^{(0)}\|_2^2$;
2. *w.p. at least* $1 - 2e^{-(m-n)/32}$, $L(\boldsymbol{\theta}^{(t)}) \geq \lambda_{\min}(\boldsymbol{\Sigma}) \cdot (1 - \eta\lambda)^{2t} \cdot \frac{(m-n)\nu^2}{2}$;
3. $\|\boldsymbol{\theta}^{(t)}\|_2 \leq (1 - \eta\lambda)^t \cdot \|\boldsymbol{\theta}^{(0)}\|_2$.

*Proof of Theorem A.1.* The proof of Theorem A.1 follows from the following Theorem A.2, Theorem A.4 and Theorem A.5.

$\square$

**Theorem A.2** (**Training loss convergence**). *For all $t \in \mathbb{N}$, we have*

$$L_n(\boldsymbol{\theta}^{(t)}) \leq \frac{L}{2} \cdot \left(1 - \frac{\eta\lambda_{\min}^+(\boldsymbol{\Phi}^\top\boldsymbol{\Phi})}{n} - \eta\lambda\right)^{2t} \cdot \|\boldsymbol{\theta}^{(0)}\|_2^2.$$

**Remark A.3.** *Let $\boldsymbol{\Gamma} := \frac{1}{n}\boldsymbol{\Phi}^\top\boldsymbol{\Phi} + \lambda\boldsymbol{I}_m$ and let $\lambda_{\min}(\boldsymbol{\Gamma})$ denote its smallest eigenvalue. Following a standard argument for showing convergence for ridge regression, we can easily prove $L_n(\boldsymbol{\theta}^{(t)}) \leq L(1 - \eta\lambda_{\min}(\boldsymbol{\Gamma}))^{2t}\|\boldsymbol{\theta}^{(0)}\|_2^2$. To show grokking, we need to prove that the convergence of the training error is faster than the generalization error. However, when the feature map $\boldsymbol{\Phi}$ is not full rank, $\lambda_{\min}(\boldsymbol{\Gamma}) = \lambda$, making this rate match the lower bound in Theorem A.4 below, and thus does not suffice to prove grokking. Therefore, a more tight bound on the convergence rate is required here.*

*Proof of Theorem A.2.* For any vector $\boldsymbol{\theta} \in \mathbb{R}^m$, we write the following unique decomposition $\boldsymbol{\theta} = \boldsymbol{\theta}_\parallel + \boldsymbol{\theta}_\perp$ where $\boldsymbol{\theta}_\parallel$ is in the row space of $\boldsymbol{\Phi}$ and $\boldsymbol{\theta}_\perp$ is orthogonal to the row space of $\boldsymbol{\Phi}$ (or equivalently, $\boldsymbol{\theta}_\perp$ is in the null space of $\boldsymbol{\Phi}$). We can express the training loss (at $\boldsymbol{\theta}$)

$$L_n(\boldsymbol{\theta}) = \frac{1}{2n}\sum_{i=1}^n (\langle\boldsymbol{\theta}, \boldsymbol{\phi}(\boldsymbol{x}_i)\rangle)^2 = \frac{1}{2n}\sum_{i=1}^n (\langle\boldsymbol{\theta}_\parallel, \boldsymbol{\phi}(\boldsymbol{x}_i)\rangle)^2 = L_n(\boldsymbol{\theta}_\parallel)$$

as a function of $\boldsymbol{\theta}_\parallel$. Note that

$$
\begin{aligned}
\|\nabla_{\boldsymbol{\theta}}L_n(\boldsymbol{\theta}) - \nabla_{\boldsymbol{\theta}}L_n(\boldsymbol{\theta}')\|_2 &= \left\|\frac{1}{n}\sum_{i=1}^n N(\boldsymbol{x}_i; \boldsymbol{\theta})\boldsymbol{\phi}(\boldsymbol{x}_i) - \frac{1}{n}\sum_{i=1}^n N(\boldsymbol{x}_i; \boldsymbol{\theta}')\boldsymbol{\phi}(\boldsymbol{x}_i)\right\|_2 \\
&= \left\|\frac{1}{n}\sum_{i=1}^n \boldsymbol{\phi}(\boldsymbol{x}_i)\langle\boldsymbol{\theta} - \boldsymbol{\theta}', \boldsymbol{\phi}(\boldsymbol{x}_i)\rangle\right\|_2 \\
&\leq \sqrt{\left(\frac{1}{n}\sum_{i=1}^n (\langle\boldsymbol{\theta} - \boldsymbol{\theta}', \boldsymbol{\phi}(\boldsymbol{x}_i)\rangle)^2\right)\left(\frac{1}{n}\sum_{i=1}^n \|\boldsymbol{\phi}(\boldsymbol{x}_i)\|_2^2\right)} \\
&\leq \left(\frac{1}{n}\sum_{i=1}^n \|\boldsymbol{\phi}(\boldsymbol{x}_i)\|_2^2\right) \cdot \|\boldsymbol{\theta} - \boldsymbol{\theta}'\|_2 = L\|\boldsymbol{\theta} - \boldsymbol{\theta}'\|_2.
\end{aligned}
$$

Due to the above Lipschitz continuity of the gradient, we have

$$
\begin{aligned}
L_n(\boldsymbol{\theta}^{(t)}) - L_n(\boldsymbol{\theta}^*) &= L_n(\boldsymbol{\theta}_\parallel^{(t)}) - L_n(\boldsymbol{\theta}_\parallel^*) \\
&\leq \left(\nabla_{\boldsymbol{\theta}}L_n(\boldsymbol{\theta}_\parallel^*)\right)^\top \left(\boldsymbol{\theta}_\parallel^{(t)} - \boldsymbol{\theta}_\parallel^*\right) + \frac{L}{2}\left\|\boldsymbol{\theta}_\parallel^{(t)} - \boldsymbol{\theta}_\parallel^*\right\|_2^2 = \frac{L}{2}\left\|\boldsymbol{\theta}_\parallel^{(t)} - \boldsymbol{\theta}_\parallel^*\right\|_2^2. \quad (9)
\end{aligned}
$$

Next, we write out the GD iteration as

$$
\begin{aligned}
\boldsymbol{\theta}^{(t)} &= \boldsymbol{\theta}^{(t-1)} - \frac{\eta}{n} \sum_{i=1}^{n} N(\boldsymbol{x}_i; \boldsymbol{\theta}^{(t-1)}) \boldsymbol{\phi}(\boldsymbol{x}_i) - \eta\lambda\boldsymbol{\theta}^{(t-1)} \\
&= \boldsymbol{\theta}^{(t-1)} - \frac{\eta}{n} \sum_{i=1}^{n} \left\langle \boldsymbol{\theta}^{(t-1)}, \boldsymbol{\phi}(\boldsymbol{x}_i) \right\rangle \boldsymbol{\phi}(\boldsymbol{x}_i) - \eta\lambda\boldsymbol{\theta}^{(t-1)} \\
&= \left( \boldsymbol{I}_m - \eta \left( \frac{1}{n}\boldsymbol{\Phi}^\top\boldsymbol{\Phi} + \lambda\boldsymbol{I}_m \right) \right) \boldsymbol{\theta}^{(t-1)}.
\end{aligned}
\tag{10}
$$

Based on the orthogonal decomposition, we can further write

$$
\begin{aligned}
\boldsymbol{\theta}_\|^{(t+1)} &= \boldsymbol{\theta}_\|^{(t)} - \eta\boldsymbol{\Phi}^\top\boldsymbol{\Phi}\boldsymbol{\theta}_\|^{(t)} - \eta\lambda\boldsymbol{\theta}_\|^{(t)}, \\
\boldsymbol{\theta}_\perp^{(t+1)} &= \boldsymbol{\theta}_\perp^{(t)} - \eta\lambda\boldsymbol{\theta}_\perp^{(t)} = (1 - \eta\lambda)\boldsymbol{\theta}_\perp^{(t)}.
\end{aligned}
$$

Let $\lambda_{\min}^+(\boldsymbol{\Phi}^\top\boldsymbol{\Phi})$ denote the smallest non-zero eigenvalue of the matrix $\boldsymbol{\Phi}^\top\boldsymbol{\Phi}$. It follows that

$$
\|\boldsymbol{\theta}_\|^{(t)}\|_2^2 = \left\| \left( \boldsymbol{I}_m - \frac{\eta}{n}\boldsymbol{\Phi}^\top\boldsymbol{\Phi} - \eta\lambda\boldsymbol{I}_m \right)^t \boldsymbol{\theta}_\|^{(0)} \right\|_2^2 \le \left( 1 - \frac{\eta\lambda_{\min}^+(\boldsymbol{\Phi}^\top\boldsymbol{\Phi})}{n} - \eta\lambda \right)^{2t} \|\boldsymbol{\theta}_\|^{(0)}\|_2^2.
\tag{11}
$$

To have the above hold, it suffices to choose a sufficiently small step size

$$
\eta < \frac{1}{\lambda + (1/n)\lambda_{\max}(\boldsymbol{\Phi}^\top\boldsymbol{\Phi})}.
$$

Since $\boldsymbol{\theta}^* = \boldsymbol{0}$ and $L_n(\boldsymbol{\theta}^*) = 0$, we have from Equations (9) and (11) that

$$
L_n\left(\boldsymbol{\theta}^{(t)}\right) \le \frac{L}{2}\|\boldsymbol{\theta}_\|^{(t)}\|_2^2 \le \frac{L}{2} \cdot \left( 1 - \frac{\eta\lambda_{\min}^+(\boldsymbol{\Phi}^\top\boldsymbol{\Phi})}{n} - \eta\lambda \right)^{2t} \cdot \|\boldsymbol{\theta}_\|^{(0)}\|_2^2.
$$

Note that the convergence rate of $(1 - \eta\lambda_{\min}^+(\boldsymbol{\Phi}^\top\boldsymbol{\Phi})/n - \eta\lambda/n)^{2t}$ is always faster than $(1 - \eta\lambda/n)^{2t}$. In particular, a small weight decay $\lambda$ will amplify grokking. The proof is done! $\square$

**Theorem A.4 (Overfitting).** *Assume that* $\boldsymbol{\theta}^{(0)} \sim \mathcal{N}(\boldsymbol{0}, \nu^2\boldsymbol{I}_m)$. *Then, for all* $t \in \mathbb{N}$, *we have*

$$
L(\boldsymbol{\theta}^{(t)}) \ge \lambda_{\min}(\boldsymbol{\Sigma}) \cdot (1 - \eta\lambda)^{2t} \cdot \frac{(m-n)\nu^2}{2},
$$

*with probability at least* $1 - 2e^{-(m-n)/32}$.

*Proof of Theorem A.4.* By definition, the generalization error satisfies

$$
\mathbb{E}_{\boldsymbol{x}}\left[ \left( N(\boldsymbol{x}; \boldsymbol{\theta}^{(t)}) - N^*(\boldsymbol{x}) \right)^2 \right] = (\boldsymbol{\theta}^{(t)})^\top \boldsymbol{\Sigma}\boldsymbol{\theta}^{(t)} \ge \lambda_{\min}(\boldsymbol{\Sigma}) \cdot \|\boldsymbol{\theta}^{(t)}\|_2^2.
$$

Recall that $\boldsymbol{\Phi} := (\boldsymbol{\phi}(\boldsymbol{x}_1), \ldots, \boldsymbol{\phi}(\boldsymbol{x}_n))^\top$ and $\boldsymbol{\theta}^{(t+1)} = \boldsymbol{\theta}^{(t)} - (\eta/n)\boldsymbol{\Phi}^\top\boldsymbol{\Phi}\boldsymbol{\theta}^{(t)} - \eta\lambda\boldsymbol{\theta}^{(t)}$. For any $\boldsymbol{\theta} \in \mathbb{R}^m$, we inherit the notations $\boldsymbol{\theta} = \boldsymbol{\theta}_\| + \boldsymbol{\theta}_\perp$ used in the proof of Theorem A.2. It holds that

$$
\begin{aligned}
\boldsymbol{\theta}_\|^{(t+1)} &= \boldsymbol{\theta}_\|^{(t)} - (\eta/n)\boldsymbol{\Phi}^\top\boldsymbol{\Phi}\boldsymbol{\theta}_\|^{(t)} - \eta\lambda\boldsymbol{\theta}_\|^{(t)}, \\
\boldsymbol{\theta}_\perp^{(t+1)} &= \boldsymbol{\theta}_\perp^{(t)} - \eta\lambda\boldsymbol{\theta}_\perp^{(t)} = (1 - \eta\lambda)\boldsymbol{\theta}_\perp^{(t)}.
\end{aligned}
$$

Hence, we have at step $t > 0$ that

$$
\boldsymbol{\theta}_\perp^{(t)} = (1 - \eta\lambda)\boldsymbol{\theta}_\perp^{(t-1)} = \cdots = (1 - \eta\lambda)^t \boldsymbol{\theta}_\perp^{(0)}.
$$

Note that $\boldsymbol{\theta}_{\perp}^{(0)}$ lies in a subspace of dimensionality $k$, which is at least $m - n$. Let $\boldsymbol{e}_1, \ldots, \boldsymbol{e}_k$ represent an orthogonal basis for the subspace, we have

$$\|\boldsymbol{\theta}_{\perp}^{(0)}\|_2^2 = \sum_{j=1}^{k} \left( \langle \boldsymbol{\theta}^{(0)}, \boldsymbol{e}_j \rangle \right)^2.$$

Since $\boldsymbol{\theta}^{(0)} \sim \mathcal{N}(\mathbf{0}, \nu^2 \boldsymbol{I}_d)$, we have that $\langle \boldsymbol{\theta}^{(0)}, \boldsymbol{e}_j \rangle, j \in [k]$ are independent $\mathcal{N}(0, \nu^2)$ random variables. By standard concentration inequality for Chi-squared distribution (c.f. Dudeja & Hsu, 2018, Fact 5), we have that with probability at least $1 - 2 \exp(-k/32)$,

$$\|\boldsymbol{\theta}_{\perp}^{(0)}\|_2^2 = \sum_{j=1}^{k} \left( \langle \boldsymbol{\theta}^{(0)}, \boldsymbol{e}_j \rangle \right)^2 \geq \frac{k\nu^2}{2}.$$

Since $k \geq m - n$, we have with probability of at least $1 - 2 \exp(-(m-n)/32)$,

$$\|\boldsymbol{\theta}^{(t)}\|_2^2 \geq \|\boldsymbol{\theta}_{\perp}^{(t)}\|_2^2 = (1 - \eta\lambda)^{2t} \|\boldsymbol{\theta}_{\perp}^{(0)}\|_2^2 \geq (1 - \eta\lambda)^{2t} \frac{(m-n)\nu^2}{2}.$$

It follows that

$$\mathbb{E}_{\boldsymbol{x}} \left[ \left( N(\boldsymbol{x}; \boldsymbol{\theta}^{(t)}) - N^*(\boldsymbol{x}) \right)^2 \right] \geq \lambda_{\min}(\boldsymbol{\Sigma}) \cdot (1 - \eta\lambda)^{2t} \cdot \frac{(m-n)\nu^2}{2},$$

with probability at least $1 - 2e^{-(m-n)/32}$. $\qquad \square$

**Theorem A.5 (Generalization).** *Assume that $0 < \eta \leq 2/(L + 2\lambda)$. Then, for all $t \in \mathbb{N}$, we have*

$$\|\boldsymbol{\theta}^{(t)}\|_2 \leq (1 - \eta\lambda)^t \cdot \|\boldsymbol{\theta}^{(0)}\|_2.$$

**Remark A.6.** *Note that a student network with zero output weights is exactly the zero function.*

*Proof of Theorem A.5.* We analyze the behavior of the regularization. Note that

$$\left\| \boldsymbol{\theta}^{(t+1)} \right\|_2^2 = \left\| \boldsymbol{\theta}^{(t)} - \eta \nabla_{\boldsymbol{\theta}} L_n(\boldsymbol{\theta}^{(t)}) - \eta\lambda \boldsymbol{\theta}^{(t)} \right\|_2^2$$
$$= (1 - \eta\lambda)^2 \left\| \boldsymbol{\theta}^{(t)} \right\|_2^2 + \eta^2 \left\| \nabla_{\boldsymbol{\theta}} L_n(\boldsymbol{\theta}^{(t)}) \right\|_2^2 - 2\eta(1 - \eta\lambda) \left\langle \nabla_{\boldsymbol{\theta}} L_n(\boldsymbol{\theta}^{(t)}), \boldsymbol{\theta}^{(t)} \right\rangle,$$

and then we have

$$\left\| \boldsymbol{\theta}^{(t+1)} \right\|_2^2 - \left\| \boldsymbol{\theta}^{(t)} \right\|_2^2 - \left[ (1 - \eta\lambda)^2 - 1 \right] \left\| \boldsymbol{\theta}^{(t)} \right\|_2^2$$
$$= \left\| \boldsymbol{\theta}^{(t+1)} \right\|_2^2 - (1 - \eta\lambda)^2 \left\| \boldsymbol{\theta}^{(t)} \right\|_2^2$$
$$= \eta^2 \left\| \nabla_{\boldsymbol{\theta}} L_n(\boldsymbol{\theta}^{(t)}) \right\|_2^2 - 2\eta(1 - \eta\lambda) \left\langle \nabla_{\boldsymbol{\theta}} L_n(\boldsymbol{\theta}^{(t)}), \boldsymbol{\theta}^{(t)} \right\rangle$$
$$= \eta^2 \left\| \frac{1}{n} \sum_{i=1}^{n} N(\boldsymbol{x}_i; \boldsymbol{\theta}^{(t)}) \boldsymbol{\phi}(\boldsymbol{x}_i) \right\|_2^2 - 2\eta(1 - \eta\lambda) \cdot \frac{1}{n} \sum_{i=1}^{n} \left( N(\boldsymbol{x}_i; \boldsymbol{\theta}^{(t)}) \right)^2$$
$$\leq \eta^2 \left( \frac{1}{n} \sum_{i=1}^{n} \left( N(\boldsymbol{x}_i; \boldsymbol{\theta}^{(t)}) \right)^2 \right) \left( \frac{1}{n} \sum_{i=1}^{n} \|\boldsymbol{\phi}(\boldsymbol{x}_i)\|_2^2 \right) - 2\eta(1 - \eta\lambda) \cdot \frac{1}{n} \sum_{i=1}^{n} \left( N(\boldsymbol{x}_i; \boldsymbol{\theta}^{(t)}) \right)^2.$$

It is clear that when

$$0 < \eta \leq \frac{2}{\frac{1}{n} \sum_{i=1}^{n} \|\boldsymbol{\phi}(\boldsymbol{x}_i)\|_2^2 + 2\lambda} = \frac{2}{L + 2\lambda},$$

the total norm of parameters is decreasing in a linear rate:

$$\left\| \boldsymbol{\theta}^{(t)} \right\|_2 \leq (1 - \eta\lambda)^t \cdot \left\| \boldsymbol{\theta}^{(0)} \right\|_2.$$

This implies that GD eventually reaches $\boldsymbol{\theta}^* = \mathbf{0}$ and yields generalization. $\qquad \square$

**Theorem A.7** (**Theorem 4.2** restated). *Assume a realizable teacher function $N^*(\boldsymbol{x}) = \langle \boldsymbol{\theta}^*, \boldsymbol{\phi}(\boldsymbol{x}) \rangle$ for some $\boldsymbol{\theta}^* \in \mathbb{R}^m$. Assume the boundedness of the feature map, i.e. $\|\boldsymbol{\phi}(\boldsymbol{x})\|_2 \le b$ for any $\boldsymbol{x}$ and some constant $b > 0$. Randomly initialize a student linear regression model $N(\boldsymbol{x}; \boldsymbol{\theta}^{(0)}) = \langle \boldsymbol{\theta}^{(0)}, \boldsymbol{\phi}(\boldsymbol{x}) \rangle$ with $\boldsymbol{\theta}^{(0)} \sim \mathcal{N}(\mathbf{0}, \nu^2 \boldsymbol{I}_m)$. Consider training the student to learn the teacher using gradient descent on the ridge regression objective in Equation (1). Let $c \ge \epsilon$ be a constant. For any $\epsilon > 0$, define*

$$t_1 := t_1(\epsilon) := \max\left(\left\{t \in \mathbb{N} : L_n(\boldsymbol{\theta}^{(t)}) \ge \epsilon\right\}\right) \quad \text{and} \quad t_2 := t_2(c) := \min\left(\left\{t \in \mathbb{N} : L(\boldsymbol{\theta}^{(t)}) \le c\right\}\right).$$

*For any $\delta \in (0, 1)$, assume a sufficiently large sample size:*

$$n = \Omega\left(\frac{b^4 \|\boldsymbol{\theta}^*\|_2^4}{\epsilon^2} \log\left(\frac{1}{\delta}\right)\right),$$

*a sufficiently wide regression model:*

$$m = n + \Omega\left(\max\left\{\log\left(\frac{1}{\delta}\right), \frac{\|\boldsymbol{\theta}^*\|_2^2}{\nu^2}, \frac{c^2}{\lambda_{\min}^2(\boldsymbol{\Sigma})\nu^2}\right\}\right),$$

*and finally a sufficiently small weight decay:*

$$\lambda = O\left(\min\left\{\frac{\epsilon}{\|\boldsymbol{\theta}^*\|_2^2}, \frac{\sqrt{L}\epsilon}{\|\boldsymbol{\theta}^*\|_2}\right\}\right).$$

*Then, if $\eta < 1/(\lambda + b^2)$, we have with probability at least $1 - \delta$,*

$$t_1 \le \frac{n \ln\left(\frac{6b^2 \|\boldsymbol{\theta}^{(0)}\|_2^2}{\epsilon}\right)}{2\eta \lambda_{\min}^+(\boldsymbol{\Phi}^\top \boldsymbol{\Phi})}$$

*and*

$$t_2 \ge \frac{\ln\left(\frac{(m-n)\nu^2}{2}\left(\sqrt{\frac{c}{\lambda_{\min}(\boldsymbol{\Sigma})}} + \|\boldsymbol{\theta}^*\|_2\right)^{-2}\right)}{4\eta\lambda}.$$

*Moreover, $L(\boldsymbol{\theta}^{(t)}) \le \epsilon$ for sufficiently large $t$.*

*Proof of Theorem A.7.* Based on the following Theorem A.8 and Theorem A.9, we can obtain that, with probability at least $1 - \delta$,

$$t_1 \le \frac{1}{2} \log_{1 - \frac{\eta\lambda_{\min}^+(\boldsymbol{\Phi}^\top\boldsymbol{\Phi})}{n} - \eta\lambda}\left(\frac{\epsilon}{6L\|\boldsymbol{\theta}^{(0)}\|_2^2}\right),$$

and with probability at least $1 - \delta$,

$$t_2 \ge \frac{1}{2} \log_{1 - \eta\lambda}\left(\frac{2}{(m-n)\nu^2}\left(\sqrt{\frac{c}{\lambda_{\min}(\boldsymbol{\Sigma})}} + \|\boldsymbol{\theta}^*\|_2\right)^2\right).$$

We use the following Taylor expansion $\ln(1 - x) = -x - \frac{x^2}{2} - \frac{x^3}{3} - \cdots$. For a sufficiently small $\lambda > 0$, it implies that $\ln(1 - \eta\lambda) \ge -2\eta\lambda$ and also $\ln(1 - \frac{\eta\lambda_{\min}^+(\boldsymbol{\Phi}^\top\boldsymbol{\Phi})}{n}) \le -\frac{\eta\lambda_{\min}^+(\boldsymbol{\Phi}^\top\boldsymbol{\Phi})}{n}$. Now, we can further write

$$\begin{aligned}
t_1 &\le \frac{1}{2} \log_{1 - \frac{\eta\lambda_{\min}^+(\boldsymbol{\Phi}^\top\boldsymbol{\Phi})}{n} - \eta\lambda}\left(\frac{\epsilon}{6L\|\boldsymbol{\theta}^{(0)}\|_2^2}\right) \le \frac{1}{2} \log_{1 - \frac{\eta\lambda_{\min}^+(\boldsymbol{\Phi}^\top\boldsymbol{\Phi})}{n}}\left(\frac{\epsilon}{6L\|\boldsymbol{\theta}^{(0)}\|_2^2}\right) \\
&= \frac{\ln\left(\frac{\epsilon}{6L\|\boldsymbol{\theta}^{(0)}\|_2^2}\right)}{2\ln\left(1 - \frac{\eta\lambda_{\min}^+(\boldsymbol{\Phi}^\top\boldsymbol{\Phi})}{n}\right)} \le \frac{n \ln\left(\frac{6L\|\boldsymbol{\theta}^{(0)}\|_2^2}{\epsilon}\right)}{2\eta\lambda_{\min}^+(\boldsymbol{\Phi}^\top\boldsymbol{\Phi})},
\end{aligned}$$

and

$$t_2 \geq \frac{1}{2} \log_{1-\eta\lambda} \left( \frac{2}{(m-n)\nu^2} \left( \sqrt{\frac{c}{\lambda_{\min}(\mathbf{\Sigma})}} + \|\boldsymbol{\theta}^*\|_2 \right)^2 \right) = \frac{\ln\left( \frac{2}{(m-n)\nu^2} \left( \sqrt{\frac{c}{\lambda_{\min}(\mathbf{\Sigma})}} + \|\boldsymbol{\theta}^*\|_2 \right)^2 \right)}{2\ln(1-\eta\lambda)}$$

$$\geq \frac{\ln\left( \frac{(m-n)\nu^2}{2} \left( \sqrt{\frac{c}{\lambda_{\min}(\mathbf{\Sigma})}} + \|\boldsymbol{\theta}^*\|_2 \right)^{-2} \right)}{4\eta\lambda}.$$

The bounds on the grokking time are proved via bounding $L \leq b^2$. Finally, we explain the generalization guarantee. Let $\boldsymbol{\theta}_\lambda^* = \arg\min_{\boldsymbol{\theta}} \{L_n(\boldsymbol{\theta}; \lambda)\}$. Fix some $t > 0$, we have by triangle inequality that

$$L(\boldsymbol{\theta}^{(t)}) = (L(\boldsymbol{\theta}^{(t)}) - L(\boldsymbol{\theta}_\lambda^*)) + (L(\boldsymbol{\theta}_\lambda^*) - 2L_n(\boldsymbol{\theta}_\lambda^*)) + 2L_n(\boldsymbol{\theta}_\lambda^*)$$
$$\leq (L(\boldsymbol{\theta}^{(t)}) - L(\boldsymbol{\theta}_\lambda^*)) + (L(\boldsymbol{\theta}_\lambda^*) - 2L_n(\boldsymbol{\theta}_\lambda^*)) + 2L_n(\boldsymbol{\theta}_\lambda^*; \lambda).$$

Since $\boldsymbol{\theta}^{(t)}$ converges to $\boldsymbol{\theta}_\lambda^*$ (as we show in the proof of Theorem A.10), we have $L(\boldsymbol{\theta}^{(t)}) - L(\boldsymbol{\theta}_\lambda^*) \leq \epsilon/4$ for some sufficiently large $t > 0$. Applying Theorem A.10 with $\epsilon/4$ yields that $L(\boldsymbol{\theta}_\lambda^*) - 2L_n(\boldsymbol{\theta}_\lambda^*) \leq \epsilon/4$, with probability at least $1 - \delta$. Also, we can have $2L_n(\boldsymbol{\theta}_\lambda^*; \lambda) \leq \epsilon/2$ with a sufficiently small $\lambda$. Note that we proved the theorem with probability $1 - 3\delta$ here. This does not matter since in the theorem assumptions $\delta$ only appears in asymptotic notations which ignore constant factors. $\square$

**Theorem A.8 (Theorem 4.4 restated).** *Suppose that $N^*(\boldsymbol{x}) = \langle \boldsymbol{\theta}^*, \boldsymbol{\phi}(\boldsymbol{x}) \rangle$ for some $\boldsymbol{\theta}^* \in \mathbb{R}^m$. Assume that $\boldsymbol{\theta}^{(0)} \sim \mathcal{N}(\mathbf{0}, \nu^2 \boldsymbol{I}_m)$. For any $\epsilon > 0$ and $\delta \in (0,1)$, if*

$$m = \Omega\left( \max\left\{ \log\left(\frac{1}{\delta}\right), \frac{\|\boldsymbol{\theta}^*\|_2^2}{\nu^2} \right\} \right) \quad \text{and} \quad \lambda = O\left( \min\left\{ \frac{\epsilon}{\|\boldsymbol{\theta}^*\|_2^2}, \frac{\sqrt{L\epsilon}}{\|\boldsymbol{\theta}^*\|_2} \right\} \right),$$

*then, if $\eta < 1/(\lambda + \lambda_{\max}(\boldsymbol{\Phi}^\top \boldsymbol{\Phi})/n)$, when*

$$t \geq \frac{1}{2} \log_{1 - \frac{\eta\lambda_{\min}^+(\boldsymbol{\Phi}^\top \boldsymbol{\Phi})}{n} - \eta\lambda} \left( \frac{\epsilon}{6L\|\boldsymbol{\theta}^{(0)}\|_2^2} \right),$$

*we have $L_n(\boldsymbol{\theta}^{(t)}) \leq \epsilon$ with probability at least $1 - \delta$.*

*Proof of Theorem A.8.* Let $\boldsymbol{\theta}_\lambda^*$ denote the global minimum of the regularized objective. We consider the following decomposition of the unregularized squared loss objective:

$$L_n(\boldsymbol{\theta}^{(t)}) = L_n(\boldsymbol{\theta}^{(t)}) - L_n(\boldsymbol{\theta}_\lambda^*) + L_n(\boldsymbol{\theta}_\lambda^*).$$

Since $\boldsymbol{\theta}_\lambda^*$ is the global minimum, we have

$$\nabla_{\boldsymbol{\theta}} L_n(\boldsymbol{\theta}_\lambda^*; \lambda) = \frac{1}{n} \sum_{i=1}^n \left( N(\boldsymbol{x}_i; \boldsymbol{\theta}_\lambda^*) - N^*(\boldsymbol{x}_i) \right) \boldsymbol{\phi}(\boldsymbol{x}_i) + \lambda \boldsymbol{\theta}_\lambda^*$$
$$= \frac{1}{n} \sum_{i=1}^n \langle \boldsymbol{\theta}_\lambda^*, \boldsymbol{\phi}(\boldsymbol{x}_i) \rangle \boldsymbol{\phi}(\boldsymbol{x}_i) - \frac{1}{n} \sum_{i=1}^n N^*(\boldsymbol{x}_i) \boldsymbol{\phi}(\boldsymbol{x}_i) + \lambda \boldsymbol{\theta}_\lambda^*$$
$$= \left( \frac{1}{n} \boldsymbol{\Phi}^\top \boldsymbol{\Phi} + \lambda \boldsymbol{I}_m \right) \boldsymbol{\theta}_\lambda^* - \frac{1}{n} \sum_{i=1}^n N^*(\boldsymbol{x}_i) \boldsymbol{\phi}(\boldsymbol{x}_i) = \mathbf{0}.$$

Now, the gradient descent iteration can be written as

$$\boldsymbol{\theta}^{(t)} = \boldsymbol{\theta}^{(t-1)} - \frac{\eta}{n} \sum_{i=1}^{n} \left( N(\boldsymbol{x}_i; \boldsymbol{\theta}^{(t-1)}) - N^*(\boldsymbol{x}_i) \right) \boldsymbol{\phi}(\boldsymbol{x}_i) - \eta \lambda \boldsymbol{\theta}^{(t-1)}$$

$$= \boldsymbol{\theta}^{(t-1)} - \frac{\eta}{n} \sum_{i=1}^{n} \left\langle \boldsymbol{\theta}^{(t-1)}, \boldsymbol{\phi}(\boldsymbol{x}_i) \right\rangle \boldsymbol{\phi}(\boldsymbol{x}_i) + \frac{\eta}{n} \sum_{i=1}^{n} N^*(\boldsymbol{x}_i) \boldsymbol{\phi}(\boldsymbol{x}_i) - \eta \lambda \boldsymbol{\theta}^{(t-1)}$$

$$= \boldsymbol{\theta}^{(t-1)} - \eta \left( \frac{1}{n} \boldsymbol{\Phi}^\top \boldsymbol{\Phi} + \lambda \boldsymbol{I}_m \right) \boldsymbol{\theta}^{(t-1)} + \frac{\eta}{n} \sum_{i=1}^{n} N^*(\boldsymbol{x}_i) \boldsymbol{\phi}(\boldsymbol{x}_i)$$

$$= \boldsymbol{\theta}^{(t-1)} - \eta \left( \frac{1}{n} \boldsymbol{\Phi}^\top \boldsymbol{\Phi} + \lambda \boldsymbol{I}_m \right) (\boldsymbol{\theta}^{(t-1)} - \boldsymbol{\theta}^*_\lambda).$$

Let $\boldsymbol{u}^{(t)} := \boldsymbol{\theta}^{(t)} - \boldsymbol{\theta}^*_\lambda$ for any $t \geq 0$. The above implies that

$$\boldsymbol{u}^{(t)} = \boldsymbol{u}^{(t-1)} - \eta \left( \frac{1}{n} \boldsymbol{\Phi}^\top \boldsymbol{\Phi} + \lambda \boldsymbol{I}_m \right) \boldsymbol{u}^{(t-1)} = \left( \boldsymbol{I}_m - \eta \left( \frac{1}{n} \boldsymbol{\Phi}^\top \boldsymbol{\Phi} + \lambda \boldsymbol{I}_m \right) \right) \boldsymbol{u}^{(t-1)}.$$

For any vector $\boldsymbol{u} \in \mathbb{R}^m$, we can write a unique decomposition $\boldsymbol{u} = \boldsymbol{u}_\| + \boldsymbol{u}_\perp$ where $\boldsymbol{u}_\|$ is in the row space of $\boldsymbol{\Phi}$ and $\boldsymbol{u}_\perp$ is orthogonal to the row space of $\boldsymbol{\Phi}$ (or equivalently, $\boldsymbol{u}_\perp$ is in the null space of $\boldsymbol{\Phi}$). Then, we can further write the gradient descent along with the orthogonal decomposition:

$$\boldsymbol{u}_\|^{(t)} = \boldsymbol{u}_\|^{(t-1)} - \frac{\eta}{n} \boldsymbol{\Phi}^\top \boldsymbol{\Phi} \boldsymbol{u}_\|^{(t-1)} - \eta \lambda \boldsymbol{u}_\|^{(t-1)} = \left( \boldsymbol{I}_m - \eta \left( \frac{1}{n} \boldsymbol{\Phi}^\top \boldsymbol{\Phi} + \lambda \boldsymbol{I}_m \right) \right) \boldsymbol{u}_\|^{(t-1)},$$

$$\boldsymbol{u}_\perp^{(t)} = \boldsymbol{u}_\perp^{(t-1)} - \eta \lambda \boldsymbol{u}_\perp^{(t-1)} = (1 - \eta \lambda) \boldsymbol{u}_\perp^{(t-1)},$$

which implies

$$\|\boldsymbol{u}_\|^{(t)}\|_2^2 = \left\| \left( \boldsymbol{I}_m - \frac{\eta}{n} \boldsymbol{\Phi}^\top \boldsymbol{\Phi} - \eta \lambda \boldsymbol{I}_m \right)^t \boldsymbol{u}_\|^{(0)} \right\|_2^2$$

$$\leq \left( 1 - \frac{\eta \lambda_{\min}^+(\boldsymbol{\Phi}^\top \boldsymbol{\Phi})}{n} - \eta \lambda \right)^{2t} \|\boldsymbol{u}_\|^{(0)}\|_2^2 \leq \left( 1 - \frac{\eta \lambda_{\min}^+(\boldsymbol{\Phi}^\top \boldsymbol{\Phi})}{n} - \eta \lambda \right)^{2t} \|\boldsymbol{u}^{(0)}\|_2^2.$$

To have the above hold, it suffices to choose a sufficiently small step size:

$$\eta < \frac{1}{\lambda + (1/n) \lambda_{\max}(\boldsymbol{\Phi}^\top \boldsymbol{\Phi})}.$$

From here, we cannot follow the same argument as in the proof of Theorem A.2 via analyzing the Lipschitz continuity since $\nabla_{\boldsymbol{\theta}} L_n(\boldsymbol{\theta}^*_\lambda) \neq \boldsymbol{0}$. Note that for any vector $\boldsymbol{\theta} \in \mathbb{R}^m$, we can write the squared loss (at $\boldsymbol{\theta}$) as a function of $\boldsymbol{\theta}_\|$:

$$L_n(\boldsymbol{\theta}) = \frac{1}{2n} \sum_{i=1}^{n} (\langle \boldsymbol{\theta} - \boldsymbol{\theta}^*, \boldsymbol{\phi}(\boldsymbol{x}_i) \rangle)^2 = \frac{1}{2n} \sum_{i=1}^{n} \left( \left\langle \boldsymbol{\theta}_\| - \boldsymbol{\theta}^*_\|, \boldsymbol{\phi}(\boldsymbol{x}_i) \right\rangle \right)^2 =: \tilde{L}_n(\boldsymbol{\theta}_\|).$$

We can write $L_n(\boldsymbol{\theta}^{(t)}) - L_n(\boldsymbol{\theta}^*_\lambda)$ as follow:

$$L_n(\boldsymbol{\theta}^{(t)}) - L_n(\boldsymbol{\theta}^*_\lambda) = \tilde{L}_n(\boldsymbol{\theta}_\|^{(t)}) - \tilde{L}_n(\boldsymbol{\theta}^*_{\lambda,\|})$$

$$= \frac{1}{2n} \sum_{i=1}^{n} \left( \left\langle \boldsymbol{\theta}_\|^{(t)} - \boldsymbol{\theta}^*_\|, \boldsymbol{\phi}(\boldsymbol{x}_i) \right\rangle \right)^2 - \frac{1}{2n} \sum_{i=1}^{n} \left( \left\langle \boldsymbol{\theta}^*_{\lambda,\|} - \boldsymbol{\theta}^*_\|, \boldsymbol{\phi}(\boldsymbol{x}_i) \right\rangle \right)^2$$

$$= \frac{1}{2n} \sum_{i=1}^{n} \left( \left\langle \boldsymbol{\theta}_\|^{(t)} - \boldsymbol{\theta}^*_{\lambda,\|}, \boldsymbol{\phi}(\boldsymbol{x}_i) \right\rangle \right)^2 + \frac{1}{n} \sum_{i=1}^{n} \left\langle \boldsymbol{\theta}_\|^{(t)} - \boldsymbol{\theta}^*_{\lambda,\|}, \boldsymbol{\phi}(\boldsymbol{x}_i) \right\rangle \left\langle \boldsymbol{\theta}^*_{\lambda,\|} - \boldsymbol{\theta}^*_\|, \boldsymbol{\phi}(\boldsymbol{x}_i) \right\rangle.$$

Recall that $\frac{1}{n}\sum_{i=1}^{n}\langle\boldsymbol{\theta}_{\lambda}^{*},\phi(\boldsymbol{x}_i)\rangle\phi(\boldsymbol{x}_i) - \frac{1}{n}\sum_{i=1}^{n}\langle\boldsymbol{\theta}^{*},\phi(\boldsymbol{x}_i)\rangle\phi(\boldsymbol{x}_i) + \lambda\boldsymbol{\theta}_{\lambda}^{*} = \mathbf{0}$, which implies that $\frac{1}{n}\sum_{i=1}^{n}\langle\boldsymbol{\theta}_{\lambda,\|}^{*} - \boldsymbol{\theta}_{\|}^{*},\phi(\boldsymbol{x}_i)\rangle\phi(\boldsymbol{x}_i) + \lambda\boldsymbol{\theta}_{\lambda,\|}^{*} = \mathbf{0}$ and thus

$$
\begin{aligned}
&L_n(\boldsymbol{\theta}^{(t)}) - L_n(\boldsymbol{\theta}_{\lambda}^{*})\\
=&\frac{1}{2n}\sum_{i=1}^{n}\left(\left\langle\boldsymbol{\theta}_{\|}^{(t)} - \boldsymbol{\theta}_{\lambda,\|}^{*},\phi(\boldsymbol{x}_i)\right\rangle\right)^2 - \lambda\left\langle\boldsymbol{\theta}_{\|}^{(t)} - \boldsymbol{\theta}_{\lambda,\|}^{*},\boldsymbol{\theta}_{\lambda,\|}^{*}\right\rangle\\
=&\frac{1}{2n}\sum_{i=1}^{n}\left(\left\langle\boldsymbol{u}_{\|}^{(t)},\phi(\boldsymbol{x}_i)\right\rangle\right)^2 - \lambda\left\langle\boldsymbol{u}_{\|}^{(t)},\boldsymbol{\theta}_{\lambda,\|}^{*}\right\rangle\\
\leq&\frac{L}{2}\|\boldsymbol{u}_{\|}^{(t)}\|_2^2 + \lambda\|\boldsymbol{u}_{\|}^{(t)}\|_2\|\boldsymbol{\theta}_{\lambda,\|}^{*}\|_2\\
\leq&\frac{L}{2}\left(1 - \frac{\eta\lambda_{\min}^{+}(\boldsymbol{\Phi}^{\top}\boldsymbol{\Phi})}{n} - \eta\lambda\right)^{2t}\|\boldsymbol{u}^{(0)}\|_2^2 + \lambda\left(1 - \frac{\eta\lambda_{\min}^{+}(\boldsymbol{\Phi}^{\top}\boldsymbol{\Phi})}{n} - \eta\lambda\right)^{t}\|\boldsymbol{u}^{(0)}\|_2\|\boldsymbol{\theta}_{\lambda,\|}^{*}\|_2.
\end{aligned}
$$

It remains to bound $\|\boldsymbol{\theta}_{\lambda,\|}^{*}\|_2$ and $L_n(\boldsymbol{\theta}_{\lambda}^{*})$. Since $\boldsymbol{\theta}_{\lambda}^{*}$ is the global minimum of the regularized objective and $L_n(\boldsymbol{\theta}^{*}) = 0$, we have

$$
L_n(\boldsymbol{\theta}_{\lambda}^{*}) + \frac{\lambda}{2}\|\boldsymbol{\theta}_{\lambda}^{*}\|_2^2 \leq L_n(\boldsymbol{\theta}^{*}) + \frac{\lambda}{2}\|\boldsymbol{\theta}^{*}\|_2^2 = \frac{\lambda}{2}\|\boldsymbol{\theta}^{*}\|_2^2.
$$

It follows immediately that $\|\boldsymbol{\theta}_{\lambda,\|}^{*}\|_2 \leq \|\boldsymbol{\theta}_{\lambda}^{*}\|_2 \leq \|\boldsymbol{\theta}^{*}\|_2$ and $L_n(\boldsymbol{\theta}_{\lambda}^{*}) \leq \frac{\lambda}{2}\|\boldsymbol{\theta}^{*}\|_2^2$. Moreover, note that

$$
\|\boldsymbol{u}^{(0)}\|_2 = \|\boldsymbol{\theta}^{(0)} - \boldsymbol{\theta}_{\lambda}^{*}\|_2 \leq \|\boldsymbol{\theta}^{(0)}\|_2 + \|\boldsymbol{\theta}_{\lambda}^{*}\|_2 \leq \|\boldsymbol{\theta}^{(0)}\|_2 + \|\boldsymbol{\theta}^{*}\|_2 \leq 2\|\boldsymbol{\theta}^{(0)}\|_2,
$$

where the last step holds if $\|\boldsymbol{\theta}^{*}\|_2 \leq \|\boldsymbol{\theta}^{(0)}\|_2$. Since $\boldsymbol{\theta}^{(0)} \sim \mathcal{N}(\mathbf{0},\nu^2\boldsymbol{I}_m)$, when $m = \Omega(\log(1/\delta))$, we have from Lemma C.1 that $\|\boldsymbol{\theta}^{(0)}\|_2^2 = \Omega(m\nu^2)$ with probability at least $1 - \delta$. In particular, when

$$
m = \Omega\left(\max\left\{\log\left(\frac{1}{\delta}\right),\frac{\|\boldsymbol{\theta}^{*}\|_2^2}{\nu^2}\right\}\right),
$$

we have $\|\boldsymbol{\theta}^{*}\|_2 \leq \|\boldsymbol{\theta}^{(0)}\|_2$ with probability at least $1 - \delta$. Altogether, we have for any $t \in \mathbb{N}$,

$$
\begin{aligned}
L_n(\boldsymbol{\theta}^{(t)}) \leq& 2L\left(1 - \frac{\eta\lambda_{\min}^{+}(\boldsymbol{\Phi}^{\top}\boldsymbol{\Phi})}{n} - \eta\lambda\right)^{2t}\|\boldsymbol{\theta}^{(0)}\|_2^2\\
&+ 2\lambda\left(1 - \frac{\eta\lambda_{\min}^{+}(\boldsymbol{\Phi}^{\top}\boldsymbol{\Phi})}{n} - \eta\lambda\right)^{t}\|\boldsymbol{\theta}^{(0)}\|_2\|\boldsymbol{\theta}^{*}\|_2 + \frac{\lambda}{2}\|\boldsymbol{\theta}^{*}\|_2^2.
\end{aligned}
$$

with probability at least $1 - \delta$. Finally, for any $\epsilon > 0$, we have

$$
\begin{aligned}
L_n(\boldsymbol{\theta}^{(t)}) \leq& 2L\left(1 - \frac{\eta\lambda_{\min}^{+}(\boldsymbol{\Phi}^{\top}\boldsymbol{\Phi})}{n} - \eta\lambda\right)^{2t}\|\boldsymbol{\theta}^{(0)}\|_2^2\\
&+ 2\lambda\left(1 - \frac{\eta\lambda_{\min}^{+}(\boldsymbol{\Phi}^{\top}\boldsymbol{\Phi})}{n} - \eta\lambda\right)^{t}\|\boldsymbol{\theta}^{(0)}\|_2\|\boldsymbol{\theta}^{*}\|_2 + \frac{\lambda}{2}\|\boldsymbol{\theta}^{*}\|_2^2\\
\leq&\frac{\epsilon}{3} + \frac{\epsilon}{3} + \frac{\epsilon}{3} = \epsilon,
\end{aligned}
$$

with probability at least $1 - \delta$, if the following holds:

$$
\lambda \leq \min\left\{\frac{2\epsilon}{3\|\boldsymbol{\theta}^{*}\|_2^2},\frac{\sqrt{L\epsilon}}{\sqrt{6}\|\boldsymbol{\theta}^{*}\|_2}\right\}, \quad \text{and} \quad t \geq \frac{1}{2}\log_{1 - \frac{\eta\lambda_{\min}^{+}(\boldsymbol{\Phi}^{\top}\boldsymbol{\Phi})}{n} - \eta\lambda}\left(\frac{\epsilon}{6L\|\boldsymbol{\theta}^{(0)}\|_2^2}\right).
$$

$\square$

**Theorem A.9** (Theorem 4.5 restated). *Suppose that $N^{*}(\boldsymbol{x}) = \langle\boldsymbol{\theta}^{*},\phi(\boldsymbol{x})\rangle$ for some $\boldsymbol{\theta}^{*} \in \mathbb{R}^m$. Assume that $\boldsymbol{\theta}^{(0)} \sim \mathcal{N}(\mathbf{0},\nu^2\boldsymbol{I}_m)$. For any constant $c > 0$ and $\delta \in (0,1)$, if*

$$
m = n + \Omega\left(\max\left\{\log\left(\frac{1}{\delta}\right),\frac{\|\boldsymbol{\theta}^{*}\|_2^2}{\nu^2},\frac{c}{\lambda_{\min}(\boldsymbol{\Sigma})\nu^2}\right\}\right),
$$

*then, if $\eta < 1/\lambda$, when*

$$t \le \log_{1-\eta\lambda} \left( \sqrt{\frac{2}{(m-n)\nu^2}} \left( \sqrt{\frac{c}{\lambda_{\min}(\mathbf{\Sigma})}} + \|\boldsymbol{\theta}^*\|_2 \right) \right),$$

*we have $L(\boldsymbol{\theta}^{(t)}) \ge c$ with probability at least $1 - \delta$.*

*Proof of Theorem A.9.* Let $\mathbf{\Sigma} := \mathbb{E}_{\boldsymbol{x}}[\phi(\boldsymbol{x})\phi(\boldsymbol{x})^\top]$. Since $N^*(\boldsymbol{x}) = \langle \boldsymbol{\theta}^*, \phi(\boldsymbol{x}) \rangle$, we have

$$L(\boldsymbol{\theta}^{(t)}) = \mathbb{E}_{\boldsymbol{x}} \left[ (N(\boldsymbol{x}; \boldsymbol{\theta}^{(t)}) - N^*(\boldsymbol{x}))^2 \right] = (\boldsymbol{\theta}^{(t)} - \boldsymbol{\theta}^*)^\top \mathbf{\Sigma} (\boldsymbol{\theta}^{(t)} - \boldsymbol{\theta}^*) \ge \lambda_{\min}(\mathbf{\Sigma}) \cdot \|\boldsymbol{\theta}^{(t)} - \boldsymbol{\theta}^*\|_2^2.$$

Moreover, the gradient descent follows as

$$\begin{aligned} \boldsymbol{\theta}^{(t)} =& \boldsymbol{\theta}^{(t-1)} - \frac{\eta}{n} \sum_{i=1}^n \left( N(\boldsymbol{x}_i; \boldsymbol{\theta}^{(t-1)}) - N^*(\boldsymbol{x}_i) \right) \phi(\boldsymbol{x}_i) - \eta\lambda\boldsymbol{\theta}^{(t-1)} \\ =& \boldsymbol{\theta}^{(t-1)} - \frac{\eta}{n} \sum_{i=1}^n \left\langle \boldsymbol{\theta}^{(t-1)}, \phi(\boldsymbol{x}_i) \right\rangle \phi(\boldsymbol{x}_i) + \frac{\eta}{n} \sum_{i=1}^n N^*(\boldsymbol{x}_i)\phi(\boldsymbol{x}_i) - \eta\lambda\boldsymbol{\theta}^{(t-1)} \\ =& \boldsymbol{\theta}^{(t-1)} - \frac{\eta}{n} \mathbf{\Phi}^\top\mathbf{\Phi}\boldsymbol{\theta}^{(t-1)} - \eta\lambda\boldsymbol{\theta}^{(t-1)} + \frac{\eta}{n} \mathbf{\Phi}^\top\mathbf{\Phi}\boldsymbol{\theta}^*. \end{aligned}$$

Using the same notation $\boldsymbol{\theta} = \boldsymbol{\theta}_\| + \boldsymbol{\theta}_\perp$ for any $\boldsymbol{\theta} \in \mathbb{R}^m$ where $\boldsymbol{\theta}_\perp$ is orthogonal to the row space of $\mathbf{\Phi}$, we can write

$$\begin{aligned} \boldsymbol{\theta}_\|^{(t+1)} =& \boldsymbol{\theta}_\|^{(t)} - (\eta/n)\mathbf{\Phi}^\top\mathbf{\Phi}\boldsymbol{\theta}_\|^{(t)} - \eta\lambda\boldsymbol{\theta}_\|^{(t)} + (\eta/n)\mathbf{\Phi}^\top\mathbf{\Phi}\boldsymbol{\theta}_\|^*, \\ \boldsymbol{\theta}_\perp^{(t+1)} =& \boldsymbol{\theta}_\perp^{(t)} - \eta\lambda\boldsymbol{\theta}_\perp^{(t)} = (1 - \eta\lambda)\, \boldsymbol{\theta}_\perp^{(t)}. \end{aligned}$$

Hence, we have at step $t > 0$ that

$$\boldsymbol{\theta}_\perp^{(t)} = (1 - \eta\lambda)\, \boldsymbol{\theta}_\perp^{(t-1)} = \cdots = (1 - \eta\lambda)^t\, \boldsymbol{\theta}_\perp^{(0)}.$$

By triangle inequality, we have

$$\|\boldsymbol{\theta}^{(t)} - \boldsymbol{\theta}^*\|_2 \ge \|\boldsymbol{\theta}^{(t)}\|_2 - \|\boldsymbol{\theta}^*\|_2 \ge \|\boldsymbol{\theta}_\perp^{(t)}\|_2 - \|\boldsymbol{\theta}^*\|_2 = (1 - \eta\lambda)^t \|\boldsymbol{\theta}_\perp^{(0)}\|_2 - \|\boldsymbol{\theta}^*\|_2.$$

When $\boldsymbol{\theta}^{(0)} \sim \mathcal{N}(\mathbf{0}, \nu^2\boldsymbol{I}_d)$, following the same analysis in the proof of Theorem A.4, we get

$$\|\boldsymbol{\theta}^{(t)} - \boldsymbol{\theta}^*\|_2 \ge (1 - \eta\lambda)^t \sqrt{\frac{(m-n)\nu^2}{2}} - \|\boldsymbol{\theta}^*\|_2,$$

with probability at least $1 - 2e^{-(m-n)/32}$. When $m \ge n + 32\log(2/\delta)$, we have $2e^{-(m-n)/32} \le \delta$. This implies that with probability at least $1 - \delta$,

$$L(\boldsymbol{\theta}^{(t)}) \ge \lambda_{\min}(\mathbf{\Sigma}) \cdot \left( (1 - \eta\lambda)^t \sqrt{\frac{(m-n)\nu^2}{2}} - \|\boldsymbol{\theta}^*\|_2 \right)^2.$$

Now, for any constant $c > 0$,

$$m \ge n + \frac{8\|\boldsymbol{\theta}^*\|_2^2}{\nu^2} \quad \text{and} \quad m \ge n + \frac{8c}{\lambda_{\min}(\mathbf{\Sigma})\nu^2}$$

suffice to guarantee that $L(\boldsymbol{\theta}^{(0)}) \ge c$. Then, when

$$t \le \log_{1-\eta\lambda} \left( \sqrt{\frac{2}{(m-n)\nu^2}} \left( \sqrt{\frac{c}{\lambda_{\min}(\mathbf{\Sigma})}} + \|\boldsymbol{\theta}^*\|_2 \right) \right),$$

we have $L(\boldsymbol{\theta}^{(t)}) \ge c$, with probability at least $1 - \delta$. $\qquad\square$

**Theorem A.10** (Theorem 4.6 restated). *Suppose that $N^*(\boldsymbol{x}) = \langle \boldsymbol{\theta}^*, \boldsymbol{\phi}(\boldsymbol{x}) \rangle$ for some $\boldsymbol{\theta}^* \in \mathbb{R}^m$. Assume that $\|\boldsymbol{\phi}(\boldsymbol{x})\|_2 \leq b, \forall \boldsymbol{x}$ for some $b > 0$. For any $\epsilon > 0$ and $\delta \in (0,1)$, if*

$$n = \Omega \left( \frac{b^4 \|\boldsymbol{\theta}^*\|_2^4}{\epsilon^2} \log \left( \frac{1}{\delta} \right) \right),$$

*then, if $\eta \leq 1/(\lambda + b^2)$, we have with probability at least $1 - \delta$,*

$$L(\boldsymbol{\theta}_\lambda^*) \leq 2L_n(\boldsymbol{\theta}_\lambda^*) + \epsilon.$$

*Proof of Theorem A.10.* The proof of generalization follows from the standard argument of uniform convergence based on Rademacher complexity, which is defined formally as

**Definition A.11** (**Rademacher Complexity**). Let $S_n = \{(\boldsymbol{x}_i, y_i)\}_{i=1}^n$ be a dataset and let $\mathcal{H}$ be a function class. The (empirical) Rademacher complexity of $\mathcal{H}$ with respect to $S_n$ is defined as follow

$$\mathrm{Rad}_{S_n}(\mathcal{H}) := \mathbb{E}_{\sigma_1,\ldots,\sigma_n \sim \mathrm{Unif}(\{\pm 1\})} \left[ \sup_{h \in \mathcal{H}} \frac{1}{n} \sum_{i=1}^n \sigma_i h(\boldsymbol{x}_i) \right], \tag{12}$$

where $\sigma_1, \ldots, \sigma_n$ are called Rademacher variables.

Let us first define the following class of regression models for any positive value $B$:

$$\mathcal{H}_{\boldsymbol{\theta}}(B) := \{\boldsymbol{x} \mapsto \langle \boldsymbol{\theta}, \boldsymbol{\phi}(\boldsymbol{x}) \rangle : \|\boldsymbol{\theta}\|_2 \leq B \}.$$

The (empirical) Rademacher complexity of $\mathcal{H}_{\boldsymbol{\theta}}(B)$ can be bounded as shown in the following lemma.

**Lemma A.12.** *We have $\mathrm{Rad}_{S_n}(\mathcal{H}_{\boldsymbol{\theta}}(B)) \leq B\sqrt{L/n}$ where $L := \frac{1}{n} \sum_{i=1}^n \|\boldsymbol{\phi}(\boldsymbol{x}_i)\|_2^2$.*

*Proof of Lemma A.12.*

$$
\begin{aligned}
\mathrm{Rad}_{S_n}(\mathcal{H}_{\boldsymbol{\theta}}(B)) \quad &= \quad \frac{1}{n}\mathbb{E}\left[ \sup_{\|\boldsymbol{\theta}\|_2 \leq B} \sum_{i=1}^n \sigma_i \langle \boldsymbol{\theta}, \boldsymbol{\phi}(\boldsymbol{x}_i) \rangle \right] \\
&= \quad \frac{1}{n}\mathbb{E}\left[ \sup_{\|\boldsymbol{\theta}\|_2 \leq B} \left\langle \boldsymbol{\theta}, \sum_{i=1}^n \sigma_i \cdot \boldsymbol{\phi}(\boldsymbol{x}_i) \right\rangle \right] \\
\overset{\text{Cauchy Schwarz}}{\leq} \quad &\frac{1}{n}\mathbb{E}\left[ \sup_{\|\boldsymbol{\theta}\|_2 \leq B} \|\boldsymbol{\theta}\|_2 \left\| \sum_{i=1}^n \sigma_i \cdot \boldsymbol{\phi}(\boldsymbol{x}_i) \right\|_2 \right] \\
&\leq \quad \frac{B}{n}\mathbb{E}\left[ \sqrt{\left\| \sum_{i=1}^n \sigma_i \cdot \boldsymbol{\phi}(\boldsymbol{x}_i) \right\|_2^2} \right] \\
\overset{\text{Jensen's ineq.}}{\leq} \quad &\frac{B}{n}\sqrt{\mathbb{E}\left[ \left\| \sum_{i=1}^n \sigma_i \cdot \boldsymbol{\phi}(\boldsymbol{x}_i) \right\|_2^2 \right]} \\
&\leq \quad \frac{B}{n}\sqrt{\mathbb{E}\left[ \sum_{i=1}^n \|\sigma_i \cdot \boldsymbol{\phi}(\boldsymbol{x}_i)\|_2^2 \right]} \\
\overset{\sigma_i \in \{\pm 1\}}{=} \quad &\frac{B}{n}\sqrt{\sum_{i=1}^n \|\boldsymbol{\phi}(\boldsymbol{x}_i)\|_2^2} = B\sqrt{\frac{L}{n}}.
\end{aligned}
$$

$\square$

Next, we introduce the following classical generalization bound based on the Rademacher complexity. The theorem has been adjusted for the purpose of our setting.

**Lemma A.13** (Shalev-Shwartz & Ben-David, 2014, **Theorem 26.5**). *Let $\mathcal{H}_{\boldsymbol{\theta}}$ be a function class and $S_n = \{(\boldsymbol{x}_i, y_i)\}_{i=1}^n$ be a dataset independently selected according to some probability measure $P$. For any $N(\boldsymbol{x}; \boldsymbol{\theta}) \in \mathcal{H}_{\boldsymbol{\theta}}$, let*

$$\mathcal{L}_n(\boldsymbol{\theta}) := \frac{1}{n} \sum_{i=1}^n \ell(N(\boldsymbol{x}_i; \boldsymbol{\theta}), y_i) \ \text{and} \ \mathcal{L}_P(\boldsymbol{\theta}) := \mathbb{E}_{(\boldsymbol{x},y)\sim P}[\ell(N(\boldsymbol{x}; \boldsymbol{\theta}), y)]$$

*with some loss function $\ell(\cdot, \cdot)$. Assume the boundedness of the loss function, i.e. $|\ell(\cdot, \cdot)| \leq c$. Then, for any integer $n > 0$ and any $\delta \in (0, 1)$, we have that with probability of at least $1 - \delta$ over samples $S_n$,*

$$\mathcal{L}_P(\boldsymbol{\theta}) \leq \mathcal{L}_n(\boldsymbol{\theta}) + 2\mathrm{Rad}_{S_n}(\ell \circ \mathcal{H}_{\boldsymbol{\theta}}) + 4c\sqrt{\frac{2\log(4/\delta)}{n}},$$

*where $\ell \circ \mathcal{H} := \{(x, y) \mapsto \ell(h(x), y) : h \in \mathcal{H}\}$. Specifically, if the loss function $\ell(\cdot, \cdot)$ is $\mathcal{L}$-Lipschitz in the first argument, we can have $\mathrm{Rad}_{S_n}(\ell \circ \mathcal{H}) = O(\mathcal{L} \cdot \mathrm{Rad}_{S_n}(\mathcal{H}))$ (c.f. Theorem 12 Bartlett & Mendelson, 2002).*

With all these technical tools in hand, we now continue to prove Theorem A.10. Note that our kernel ridge regression objective is convex

$$\min_{\boldsymbol{\theta}} \{L_n(\boldsymbol{\theta}; \lambda)\} = \min_{\boldsymbol{\theta}} \left\{ \frac{1}{2n} \sum_{i=1}^n (N(\boldsymbol{x}_i; \boldsymbol{\theta}) - N^*(\boldsymbol{x}_i))^2 + \frac{\lambda}{2} \|\boldsymbol{\theta}\|_2^2 \right\}.$$

Moreover, since

$$\nabla_{\boldsymbol{\theta}} L_n(\boldsymbol{\theta}; \lambda) = \frac{1}{n} \sum_{i=1}^n \langle \boldsymbol{\theta}, \boldsymbol{\phi}(\boldsymbol{x}_i) \rangle \, \boldsymbol{\phi}(\boldsymbol{x}_i) - \frac{1}{n} \sum_{i=1}^n N^*(\boldsymbol{x}_i) \boldsymbol{\phi}(\boldsymbol{x}_i) + \lambda \boldsymbol{\theta},$$

we have by triangle inequality that

$$\|\nabla_{\boldsymbol{\theta}} L_n(\boldsymbol{\theta}; \lambda) - \nabla_{\boldsymbol{\theta}} L_n(\boldsymbol{\theta}'; \lambda)\|_2 \leq (b^2 + \lambda)\|\boldsymbol{\theta} - \boldsymbol{\theta}'\|_2,$$

i.e., the loss objective is smooth. Hence, we know that GD converges to the global minimum $\boldsymbol{\theta}_\lambda^*$ of the regularized loss objective if using a small enough step size $\eta < 2/(b^2 + \lambda)$. Moreover, since $L_n(\boldsymbol{\theta}_\lambda^*; \lambda) \leq L_n(\boldsymbol{\theta}^*; \lambda)$ and $L_n(\boldsymbol{\theta}^*) = 0$, we have $\frac{\lambda}{2}\|\boldsymbol{\theta}_\lambda^*\|_2^2 \leq L_n(\boldsymbol{\theta}_\lambda^*) + \frac{\lambda}{2}\|\boldsymbol{\theta}_\lambda^*\|_2^2 = \frac{\lambda}{2}\|\boldsymbol{\theta}^*\|_2^2$, i.e., $\|\boldsymbol{\theta}_\lambda^*\|_2 \leq \|\boldsymbol{\theta}^*\|_2$. Now, we know $N(\boldsymbol{x}_i; \boldsymbol{\theta}_\lambda^*) \in \mathcal{H}_{\boldsymbol{\theta}}(\|\boldsymbol{\theta}^*\|_2)$.

Note that, in order to apply Lemma A.13 to $\mathcal{H}_{\boldsymbol{\theta}}(\|\boldsymbol{\theta}^*\|_2)$, we need to argue the Lipschitzness of the loss function. While the squared loss is not Lipschitz globally, it is indeed Lipschitz on a bounded domain. Since we assume that $\|\boldsymbol{\phi}(\boldsymbol{x})\|_2 \leq b$ for some universal constant $b > 0$, we have for any $N(\boldsymbol{x}; \boldsymbol{\theta}) \in \mathcal{H}_{\boldsymbol{\theta}}(\|\boldsymbol{\theta}^*\|_2)$,

$$|N(\boldsymbol{x}; \boldsymbol{\theta})| = |\langle \boldsymbol{\theta}, \boldsymbol{\phi}(\boldsymbol{x}) \rangle| \leq \|\boldsymbol{\theta}\|_2 \cdot \|\boldsymbol{\phi}(\boldsymbol{x})\|_2 \leq b\|\boldsymbol{\theta}^*\|_2.$$

Hence, the squared loss is $4b\|\boldsymbol{\theta}^*\|_2$-Lipschitz in the first argument for $\mathcal{H}_{\boldsymbol{\theta}}(\|\boldsymbol{\theta}^*\|_2)$. Next, we argue the boundedness of the squared loss objective. This is clear since for any $N(\boldsymbol{x}; \boldsymbol{\theta}) \in \mathcal{H}_{\boldsymbol{\theta}}(\|\boldsymbol{\theta}^*\|_2)$, we have

$$|\ell(N(\boldsymbol{x}; \boldsymbol{\theta}), y)| = (N(\boldsymbol{x}; \boldsymbol{\theta}) - N(\boldsymbol{x}; \boldsymbol{\theta}^*))^2 \leq 4b^2\|\boldsymbol{\theta}^*\|_2^2.$$

Additionally, we also have $L = \frac{1}{n}\sum_{i=1}^n \|\boldsymbol{\phi}(\boldsymbol{x}_i)\|_2^2 \leq b^2$.

Now, applying the uniform convergence (Lemma A.13), we get with probability at least $1 - \delta$,

$$L(\boldsymbol{\theta}_\lambda^*) \leq 2L_n(\boldsymbol{\theta}_\lambda^*) + \frac{Cb^2\|\boldsymbol{\theta}^*\|_2^2}{\sqrt{n}} + 16b^2\|\boldsymbol{\theta}^*\|_2^2\sqrt{\frac{2\log(4/\delta)}{n}},$$

where $C > 0$ is some universal constant. Finally, for any $\epsilon > 0$, if the following holds:

$$n \geq \max\left\{ \frac{4C^2 b^4 \|\boldsymbol{\theta}^*\|_2^4}{\epsilon^2}, \frac{2048 b^4 \|\boldsymbol{\theta}^*\|_2^4}{\epsilon^2} \log\left(\frac{4}{\delta}\right) \right\},$$

then we have

$$\frac{Cb^2\|\boldsymbol{\theta}^*\|_2^2}{\sqrt{n}} + 16b^2\|\boldsymbol{\theta}^*\|_2^2\sqrt{\frac{2\log(4/\delta)}{n}} \leq \frac{\epsilon}{2} + \frac{\epsilon}{2} \leq \epsilon.$$

Therefore, with probability at least $1 - \delta$, we have $L(\boldsymbol{\theta}_\lambda^*) \leq 2L_n(\boldsymbol{\theta}_\lambda^*) + \epsilon$. $\qquad\square$

**Remark A.14.** *We discuss how to derive our bounds in Equation* (8) *when assuming specific distributions over features. Specifically, under the setting and notations of Theorem 4.2, we assume further that $\phi(\boldsymbol{x}_1), \ldots, \phi(\boldsymbol{x}_n)$ are drawn i.i.d. from $\mathcal{N}(\boldsymbol{0}, \frac{1}{m}\boldsymbol{I}_m)$. For any $\epsilon > 0$, we also assume that $m = \Omega(\|\boldsymbol{\theta}^*\|_2^2/\epsilon)$. The bounds then follow directly from applying Theorem 4.2. Since $\phi(\boldsymbol{x}_1), \ldots, \phi(\boldsymbol{x}_n)$ are i.i.d. $\mathcal{N}(\boldsymbol{0}, \frac{1}{m}\boldsymbol{I}_m)$ random variables, we can have $b^2 = 3/2$ with high probability and $\lambda_{\min}(\boldsymbol{\Sigma}) = 1/m$. Moreover, since $\boldsymbol{\theta}^{(0)} \sim \mathcal{N}(\boldsymbol{0}, \nu^2 \boldsymbol{I}_m)$, we have $m\nu^2/2 \leq \|\boldsymbol{\theta}^{(0)}\|_2^2 \leq 3m\nu^2/2$ with high probability. For $t_1$, we have*

$$t_1 \leq \frac{n \ln\left(\frac{6b^2 \|\boldsymbol{\theta}^{(0)}\|_2^2}{\epsilon}\right)}{2\eta\lambda_{\min}^+(\boldsymbol{\Phi}^\top \boldsymbol{\Phi})} \leq \frac{n \ln\left(\frac{14m\nu^2}{\epsilon}\right)}{2\eta\lambda_{\min}^+(\boldsymbol{\Phi}^\top \boldsymbol{\Phi})}.$$

*For $t_2$, note that according to the assumption, we have $c = \epsilon$ and $\epsilon > \lambda_{\min}(\boldsymbol{\Sigma})\|\boldsymbol{\theta}^*\|_2^2 = \|\boldsymbol{\theta}^*\|_2^2/m$ and thus*

$$\frac{(m-n)\nu^2}{2}\left(\sqrt{\frac{c}{\lambda_{\min}(\boldsymbol{\Sigma})}} + \|\boldsymbol{\theta}^*\|_2\right)^{-2} \geq \frac{(m-n)\nu^2}{8m\epsilon}.$$

*Moreover, to get a better approximation, we note that the inequality $\ln(1 - \eta\lambda) \geq -2\eta\lambda$ used in the proof of Theorem A.7 is loose when $\lambda$ is sufficiently small. Instead, we can have $\ln(1 - \eta\lambda) \geq -1.01\eta\lambda$ when $\eta\lambda \leq 0.01$. This gives a more tight bound followed from the same analysis:*

$$t_2 \geq \frac{\ln\left(\frac{(m-n)\nu^2}{2}\left(\sqrt{\frac{c}{\lambda_{\min}(\boldsymbol{\Sigma})}} + \|\boldsymbol{\theta}^*\|_2\right)^{-2}\right)}{2.02\eta\lambda} \geq \frac{\ln\left(\frac{(m-n)\nu^2}{8m\epsilon}\right)}{2.02\eta\lambda}.$$

*Putting together, we get*

$$t_2 \geq \frac{\ln\left(\frac{(m-n)\nu^2}{8m\epsilon}\right)}{2.02\eta\lambda} \quad \text{and} \quad t_1 \leq \frac{n \ln\left(\frac{14m\nu^2}{\epsilon}\right)}{2\eta\lambda_{\min}^+(\boldsymbol{\Phi}^\top \boldsymbol{\Phi})}.$$

## B. Additional Experiments

In this section we provide several additional experiments.

First, we define a threshold-based surrogate accuracy metric by

$$\mathbb{P}_{\boldsymbol{x}}\left(\left(N(\boldsymbol{x}; \boldsymbol{\theta}^{(t)}) - N^*(\boldsymbol{x})\right)^2 \leq \epsilon\right)$$

for a fixed $\epsilon > 0$. Evaluating our model under this metric reveals the familiar plateauing phenomenon experimentally (Figure 5). This confirms that the plateau can be recovered in linear regression under appropriate evaluation, offering a promising direction for future theoretical analysis.

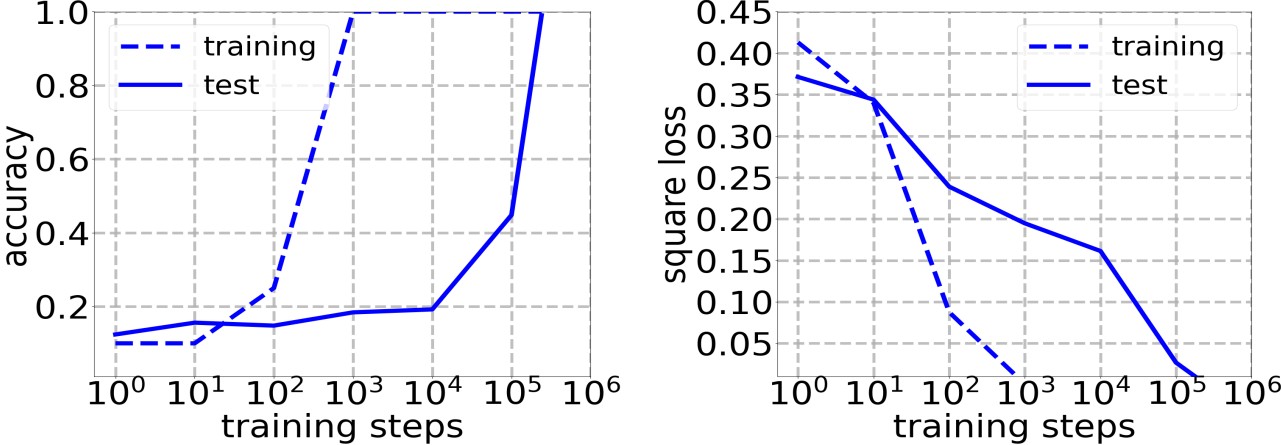

*Figure 5.* Plotting the grokking by evaluating the surrogate loss (an accuracy-type metric) compared to evaluating the actual loss. The data for the two plots are collected from a single simulation. We set $m = 10000$, $d = 100$, $n = 100$, $\eta = 1$, $\lambda = 10^{-5}$ and $\epsilon = 0.01$.

We also observe grokking in ridge regression with labels that are contaminated with mean-zero variance-std$^2$ Gaussian noise (Figure 6).

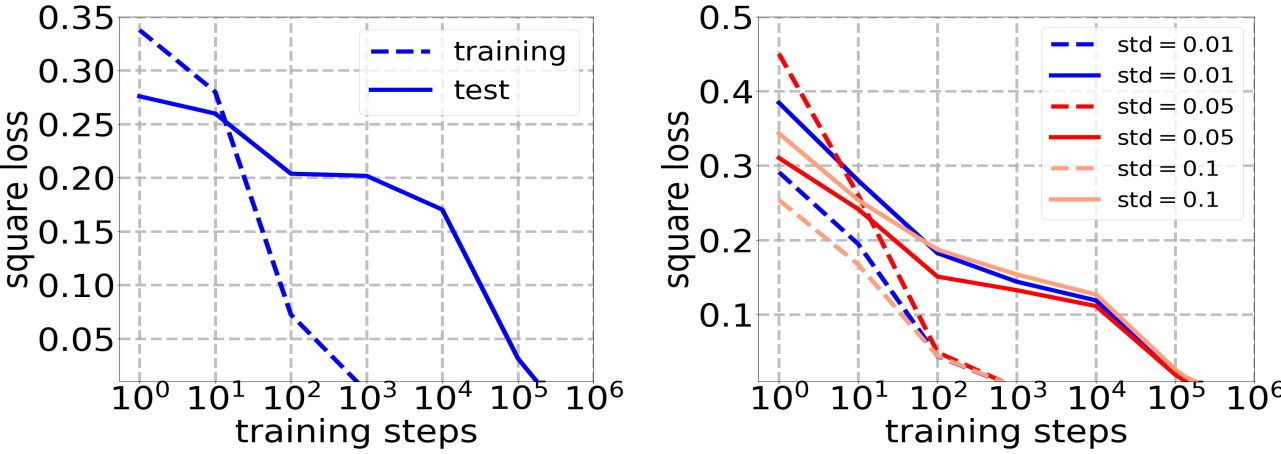

*Figure 6.* Ridge regression with label noise — **Left**: We set $m = 5000$, $d = 50$, $n = 100$, $\eta = 1$, $\mathrm{std} = 0.1$ and $\lambda = 10^{-5}$; **Right**: We notice that different noise levels do not affect the grokking time by much. We set $m = 5000$, $d = 50$, $n = 100$, $\eta = 1$, and $\lambda = 10^{-5}$.

Finally, we observe grokking in the following setting. We study training a random Fourier feature network to learn a randomly-initialized realizable teacher. We implemented the random Fourier feature map $\phi(\boldsymbol{x}) = \sqrt{2/m} \cos(\boldsymbol{W}\boldsymbol{x} + \boldsymbol{b})$ where each row of $\boldsymbol{W}$ is sampled from $\mathcal{N}(\boldsymbol{0}, \sigma^2 \boldsymbol{I})$ and each entry of $\boldsymbol{b}$ is sampled from $\mathrm{Uniform}([0, 2\pi])$. Our randomly-initialized realizable teacher is a single such Fourier feature function generated from the same distribution. We note that this setting requires careful hyperparameter tuning to observe grokking (Figure 7).

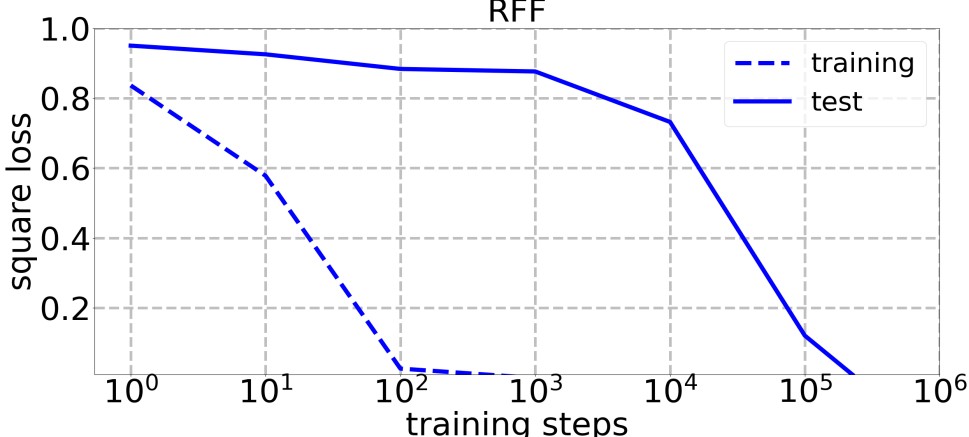

*Figure 7.* Random Fourier features — we set $m = 5000$, $d = 10$, $n = 100$, $\eta = 1$, $\sigma^2 = 1$ and $\lambda = 10^{-4}$.

## C. Technical Lemmas

**Lemma C.1 (Chi-squared Concentration).** *Let $\boldsymbol{w} \in \mathbb{R}^d$ such that $\boldsymbol{w} \sim \mathcal{N}(\boldsymbol{0}, \sigma^2 \boldsymbol{I}_d)$ for some $\sigma^2 > 0$. For all $t \in (0, 1)$,*

$$\mathbb{P}\left( \frac{1}{d\sigma^2} \left| \|\boldsymbol{w}\|_2^2 - d\sigma^2 \right| \geq t \right) \leq 2e^{-dt^2/8}.$$

