# OpenReview forum: "To Grok Grokking: Provable Grokking in Ridge Regression"
_ICML.cc/2026/Conference — ICML 2026 spotlight_

### Official Review · Reviewer_pbHK · 2026-02-24

**Soundness:** 4
**Presentation:** 3
**Significance:** 4
**Originality:** 3
**Overall Recommendation:** 6
**Confidence:** 4

**Summary:**

The authors present a novel theory to explain the phenomenon of grokking in the case of over-parametrized linear regression models using gradient descent with weight decay, based around the decay of the left nullspace of the model-data Jacobian at a potentially much slower rate than learning occurs in the range of the Jacobian. This implies that grokking is essentially an emergent property of certain training hyperparameter regimes. The authors derive quantitative bounds for the two timescales associated with grokking as a function of training hyperparameters, and show empirically that these bounds are reflected in reality. They also provide empirical evidence that grokking can also be induced in non-linear neural networks by varying analogous training hyperparameters, albeit without quantitative bounds in this setting.

**Compliance With Llm Reviewing Policy:**

Affirmed.

**Final Justification:**

My original stance was that this paper was a significant contribution to an important area of the literature, but was perhaps lacking in surrounding discussion and accuracy of claims. Following the rebuttal, which expanded on both of these areas, I am fully satisfied and advocate strongly for acceptance.

**Key Questions For Authors:**

# Claim that "the generalization error eventually becomes arbitrarily small"
The reviewer finds this claim to be much to strong, as it only holds in the overtly artificial case that a) the true function lies in the span of the feature map, and b) the exact feature coefficients correspond exactly to the weight decay target parameters, namely 0 in this work. Would the authors be able to back up a weaker version of this claim with a much more realistic scenario? For example, a random Fourier feature map and randomly-initialized realizable teacher randomly would be a compelling numerical experiment in which the generalization error would eventually become both low and predictable (via results from Gaussian Process theory). With the addition of this demonstration, the reviewer would be happy to increase their score by 1.

# Interpretation of the terminal solution in linear models
When training with gradient descent and weight decay, as the authors point out the eventual solution will be one in which the orthogonal parameter subspace is 0. This corresponds exactly to the case of training without weight decay from a 0 initialisation, rather than randomly initialized weights. Now, it is well-known that linear ridge regressions trained with gradient descent converge to the posterior mean of a Gaussian Process conditioned on the data equipped with a prior mean equal to the initialisation of the linear model and a covariance function corresponding to the kernel induced by the feature map. With this understanding, it is clear that with weight decay and in the regime after near-convergence of training loss, essentially what is changing is the prior mean of the GP for which the instantaneous solution corresponds to the posterior of, with grokking thus corresponding under the paper's theory to a transition from misspecified to well-specified prior mean. The reviewer believes discussion of this would be beneficial to the paper, and make the theory more grounded and compelling, and allow for more interpretable design of demonstrative experiments. Would the authors be willing to discuss this point in the revised manuscript? Furthermore, the reviewer is curious as to how initialisation from a particular parameter setting rather than random would affect the theory and predicted timescales?

# Interpretation of the terminal solution in non-linear models
In contrast to linear models, where weight decay causes the effective prior mean to decay to 0, research has shown that in certain cases, weight decay in wide non-linear neural networks causes the initial neural tangent kernel to decay leaving only the data-dependent NTK induced by feature learning (Lauditi et al. 2025). This maintains the view that the implicit prior associated with the learned solution is what changes during grokking. The reviewer is curious on the perspectives of the authors on this, and how the paper’s theory might apply here.

If discussion on the latter two points is included, the reviewer would be happy to further increment their score by 1.


Lauditi, C., Bordelon, B. and Pehlevan, C., Adaptive kernel predictors from feature-learning infinite limits of neural networks. In Forty-second International Conference on Machine Learning, 2025.

**Limitations:**

Yes

**Strengths And Weaknesses:**

# Strengths #
- Soundness: The theory is well-supported by rigorous proofs, and is intuitively compelling.
- Presentation: The logical flow of the paper is easy to follow, and a good overview of existing literature is present, with the paper being situated in the literature as the first thoery providing quantitative bounds for the relevant timescales of grokking. Mathematical notation in the main body is used only when necessary, so does not read as overly dense.
- Significance: The paper marks a significant step in understanding the important phenonemon of grokking, and provides the first quantitative explanation and bounds, albeit for a relatively simply case of linear models. The room for follow-up theoretical work is substantial, moving from linear models to wide neural networks in kernel and ideally feature-learning regimes.
- Originality: The underlying idea and analysis are, to the best of the reviewer's knowledge, relatively novel, well-motivated, and compelling.

# Weaknesses #
- Soundness: The reviewer finds the claim (from the abstract) that "the generalization error eventually becomes
arbitrarily small" to be much too strong. Indeed, the provided example in which this claim does hold is not compelling, for reasons which are discussed in the questions section below. Additionally, the statement "Such an imbalanced trajectory prevents timely
generalization and leads to overfitting" is a little inaccurate - overfitting is inevitable for an overparametrized linear model with sufficiently diverse features; what changes at the onset of grokking in this setting is a transition from harmful overfitting to benign overfitting.
- Presentation: Many of the figures and laterally distorted, and would benefit from larger text. Additionally, the paper would benefit from greater discussion of the theory's implications in relation to the literature. For example, other works (such as Prieto et al. 2025, referencing the paper's bibliography), have shown that grokking may arise without explicit weight decay mechanisms. Given that this paper's theory explicitly requires a weight decay mechanism, this difference should be discussed.
- Significance: Continuing from above, this theory is only able to explain weight decay-related grokking, and thus a complete understanding of all the ways in which grokking can arise is still a distant prospect. That said, this is a monumental task, and the reviewer would like to make it clear that they believe this paper is an important step in the right direction. Additionally, the theory would be more compelling and complete if incrementally further ideas were discussed, which are laid out in the questions section below.
- Originality: The idea that weight decay, or training hyperparameters in general, are strongly-contributing mechanism for grokking is not itself new, so in a way these results are expected.

---

> ### Author Rebuttal · Authors · 2026-03-30
>
> We sincerely thank the reviewer for taking the time and effort to review our paper. We appreciate their positive remarks regarding the contributions and significance of our work, as well as the quality of our presentation. Below, we provide detailed responses to your questions and concerns.
>
> W1:
> First, see our response to Q1. Second, as you pointed out, in the sentence "Such an imbalanced trajectory prevents timely generalization and leads to overfitting" we meant "harmful overfitting". We will rephrase accordingly to avoid confusion.
>
> W2:
> Thanks, we will fix the figures. Indeed, several prior works showed grokking without weight decay; we will discuss this point more explicitly in the revision.
>
> W3:
> We will add a discussion about further ideas as the reviewer suggests.
>
> W4:
> Indeed, the idea that training hyperparameters strongly affect grokking is not new. The novelty lies in our rigorous analysis and quantitative bounds.
>
> Q1:
> The sentence "the generalization error eventually becomes arbitrarily small" refers to the fact that under the assumptions of Theorem 4.2, we eventually have test error smaller than epsilon.
> As the reviewer suggested, we conducted additional experiments with a random Fourier feature map and a randomly-initialized realizable teacher. We implemented the following random Fourier feature map $\phi(\mathbf{x})=\sqrt{2/m}\cos(\mathbf{Wx}+\mathbf{b})$ where each row of $\mathbf{W}$ is sampled from $\mathcal{N}(\mathbf{0},\sigma^{2}\mathbf{I})$ and each entry of $\mathbf{b}$ is sampled from $\mathrm{Unif}([0,2\pi])$. Our randomly-initialized realizable teacher is a single such Fourier feature function generated from the same distribution. We then follow the same strategy of this paper to train a wide random feature model (independent to the teacher) to learn the teacher. We set the model hyperparameters to $m=1000$, $d=10$, $n=100$, $\eta=1$, $\sigma^{2}=1$ and $\lambda=10^{-5}$. The authors are glad to report that we did also observe grokking under this setup. Since ICML does not allow attachments in the rebuttal, we cannot include the figure here, and we will discuss it in the final version and include the newly generated figure there.
>
> Q2:
> That is an interesting perspective. We will add a short discussion regarding how grokking in our setting can be viewed as a transition from a misspecified to a well-specified prior mean, as suggested.
> Regarding a fixed initialization, it is clear that the training loss can still rapidly drop to zero. For proving overfitting, let us briefly recall that the main intuition for showing a lower bound on the test loss is that we are in a high-dimensional regression setting, i.e., having sufficiently many features and a much smaller sample size ($m\gg n$). During the training process, GD will only effectively update the regression weights projected onto the subspace spanned by data feature maps, while leaving the weights projected on the orthogonal complement subspace nearly unchanged and close to their initial values. Based on this, we can then lower bound the test loss by showing that a randomly initialized student model does not generalize well. Now, if the student is initialized deterministically, the last part of our proof still holds since we still have randomness induced by the test point. In other words, any fixed student model should not generalize well on a random test point with high probability. Finally, generalization will eventually occur with the assistance of weight decay. In our work, we focused on random initialization as it is commonly adopted in practice.
>
>
> Q3:
> We thank the reviewer for the interesting reference. Indeed, both in our analysis and in Lauditi et al., the implicit prior changes due to weight decay. This can serve as a unifying perspective for investigating grokking. In our setting, we prove that the prior change leads to grokking. It would be interesting to study under what settings the prior change from Lauditi et al. also leads to grokking, and whether one can obtain quantitative bounds on the grokking time. Currently, we are unsure whether our theory has direct implications for the setting of Lauditi et al., and we will further explore this and include a short discussion on this direction in the final version as suggested.

---

> > ### Author Rebuttal · Reviewer_pbHK · 2026-03-31
> >
> > I thank for the authors for providing detailed responses to my questions and for running the additional experiment. I enjoyed the thoughtful discussion and, given that my prior concerns have been fully addressed, will raise my score to 6 as promised.
> >
> > I'd like to restate again here that I thoroughly enjoyed reading this work, and recognise that it forms an impactful and original contribution to an important area of research. I look forward to seeing future research along these lines.

---

> > > ### Author Response · Authors · 2026-04-06
> > >
> > > Thank you for carefully reading our rebuttal and for your positive feedback. We are grateful that you found our response helpful, and appreciate your insightful comments throughout the review process.

---

### Official Review · Reviewer_MztS · 2026-03-05

**Soundness:** 3
**Presentation:** 2
**Significance:** 3
**Originality:** 3
**Overall Recommendation:** 4
**Confidence:** 3

**Summary:**

This paper studies the phenomenon of grokking in a ridge regression setting. The main result shows that, in the over-parameterized setting, the trained model initially overfits but subsequently achieves arbitrarily small test error, reflecting the characteristic lag between training and test performance of grokking. The analysis identifies key hyperparameters governing grokking, and the empirical findings corroborate the theoretical predictions.

**Compliance With Llm Reviewing Policy:**

Affirmed.

**Final Justification:**

My main concern that the addressed phenomenon did not take into account the plateauing behavior observed in grokking has been addressed by the paper. The paper is technically sound and demonstrates the time lag between training and test loss, along with insights into how the relevant parameters can be tuned. I find this theoretical contribution interesting and promising for further understanding grokking.

However, I still have doubts about the extent to which this work fully captures the phenomenon of grokking, which is of interest across a wide range of domains involving nonlinear deep models with more complex feature learning interactions. One may interpret the results as primarily demonstrating a time lag in linear models rather than capturing the full grokking phenomenon. Therefore, I would prefer to vote for a borderline **weak** accept.

**Key Questions For Authors:**

Q1. Can the theory developed by the authors account for the plateauing phenomenon of grokking?

Q2. Could the authors address my concern mentioned in W2? I believe further discussion by taking into account such t_3 may strengthen the discussion on the essence of the time lag $t_2-t_1$.

Q3. How do the theoretical results derived here relate to other theoretical/emprirical works studying grokking?

**Limitations:**

The limitations of the proposed approach are not discussed explicitly, although they are implied throughout the paper. A dedicated section summarizing and discussing these limitations would be appreciated.

**Strengths And Weaknesses:**

(Strengths)

S1 The paper is mostly well-written and easy to follow. Notably, the authors structure the argument by first analyzing a simplified setting and then extending the results to the general case. This progression is helpful for developing intuition before diving into the complex maths.

S2. The main theorems showing the different time steps are sound and clear. There is also extensive discussion to interpret these results, which I think are interesting.

S3. The experiments correctly consider and illustrate important aspects of the main points of this paper.

(Weaknesses)

W1. While the authors successfully prove the time lag between training and test errors, this only treats a partial aspect of grokking. Indeed, one of the most interesting characteristics of grokking resides in the test error that plateaus while the training performance decreases. Although the authors acknowledge this as an essential component of grokking, they do not provide theory to explain it, and only focus on the time lag. Experiments that are used to illustrate grokking do not exhibit such a behavior either. This raises some questions about whether the chosen problem setting is appropriate for studying grokking.

W2. The derivation of the time lag between training and test errors presents also an issue. The authors consider an upper bound of the empirical loss and a lower bound of the population loss. From these inqualities, they derive a lower (upper) bound on the time $t_1$ ($t_2$) required for the empirical (population) loss to reach a specified threshold. Importantly, they attribute reasons for the time lag characteristic to grokking to qualitative differences between $t_1$ and $t_2$. However, the proof of the lower bound for the population loss (i.e., Theorem A.4 and A.9) can be directly applied to the empirical loss, providing a potentially qualitatively similar time step to $t_2$ (let us define it as $t_3$). To complete the discussion, the authors should discuss this $t_3$ with respect to $t_2$.  Otherwise, the theory may not fully capture the fundamental mechanism underlying grokking, as it does not explicitly account for the essential differences between the behavior of training and test losses.

W3. The paper considers the ridge regression to analyze grokking, and generalization to non-linear practical cases is unclear. The authors provide an experiment for two-layer neural networks but this could be extended to practical scenarios where grokking is observed.

W4. From line 92 (left), the authors clarify their contributions. This discussion is somewhat redundant, as it is immediately followed by a paragraph titled “Our Contributions”. Revising these sections would improve clarity and concision.

W5. (minor) Legends in Figure 2 include some green or orange error bars, but those are not present in the actual plots.

---

> ### Author Rebuttal · Authors · 2026-03-29
>
> We sincerely thank the reviewer for taking the time and effort to review our paper. We appreciate the positive remarks regarding the clarity and presentation of our work. Below, we provide detailed responses to your questions and concerns.
>
> W1/Q1. We thank the reviewer for proposing this viewpoint. Our work is the first to prove end-to-end grokking, which we defined as a situation where the model harmfully overfits, poor generalization persists long, but eventually the generalization error becomes small. While this definition indeed does not require that the test error necessarily plateaus, we agree that if it does, then this more starkly demonstrates the surprising phenomenon of grokking where generalization eventually improves after appearing to completely stall. The plateauing situation typically appears in the grokking literature when plotting the accuracy rather than the loss in classification tasks. See, for example, Figure 1 in [Gromov, 2023], where the author plotted grokking for both the accuracy and the MSE loss. In regression tasks, it is more natural to measure the loss, as there is no explicit notion of accuracy. However, by evaluating a surrogate loss defined by $\mathbb{P}_{x}[(N(x;\theta^{(t)})-N^*(x))^{2} \leq \epsilon]$ for some fixed $\epsilon$, instead of the square loss in our plots, we do observe this plateauing phenomenon. That is, our model does exhibit the plateauing phenomemon if we define an alternative measure that corresponds to "accuracy" adapted to our regression setting. Since we cannot include figures in the rebuttal, we will add this plot in the final version and further discuss plateauing as the author suggested. Lastly, we additionally remark that as implied by our results, as the weight decay parameter approaches zero, the test loss plateaus and remains larger than a constant c for an arbitrarily long time, and hence plateauing also naturally occurs in our setting with an appropriate hyperparameter tuning, which further demonstrates the thorough control our analysis obtains in this problem setting.
>
> [Gromov, 2023]: Grokking modular arithmetic.
>
> W2/Q2. We believe that there might be a possible confusion here. If we understood correctly, the reviewer mentioned in their review that we derived a lower bound on $t_1$, and an upper bound on $t_2$, where in fact we derived the opposite (upper bound on $t_1$ and lower bound on $t_2$), that allow lower bounding $t_2-t_1$. Hence, the gap between $t_2$ and $t_1$ characterizes at least how long generalization lags behind overfitting. Moreover, we would like to point out that our proof technique in Theorem A.4 and A.9 is not applicable to the training loss. The main intuition for showing a lower bound on the test loss is that we are in a high-dimensional regression setting, i.e., having sufficiently many features and a much smaller sample size ($m\gg n$). During the training process, GD will only effectively update the regression weights projected onto the subspace spanned by data feature maps, while leaving the weights projected on the orthogonal complement subspace nearly unchanged and close to their initial values. Based on this, we can then lower bound the test loss via showing that a randomly initialized student model does not generalize well. Applying such a technique to the empirical loss will obtain time $t_3$ (as the reviewer defined) that lower bounds the empirical loss obtained, which is roughly the same as our upper bound for time $t_1$. This expected tightness follows since the entire empirical loss, which only relates to the feature maps induced by data, will be optimized effectively. Since the empirical lower bound time $t_3$ is not required to establish grokking, we are not entirely certain what the motivation of the reviewer was for bringing this up, and if we have missed a core concept in their suggestion then we apologize, and we would be happy to further clarify this if needed.
>
> W3. Proving end-to-end grokking for non-linear neural networks would definitely be a strong contribution, but we believe that obtaining a rigorous understanding of the linear case is a crucial intermediate step towards that goal. In particular, we believe that unveiling the root causes for grokking in the more simplified setting of linear regression, and obtaining a complete and thorough understanding of the separate effects of each hyperparameter on it, would prove as a stepping stone towards the more complicated settings mentioned by the reviewer (see also our response to Reviewer 39zk's similar concern).
>
> W4. Thank you for this suggestion, we will further clarify these parts in a revised version.
>
> Q3. Apart from the discussion regarding this in Section 1.1, our results uncover a deeper understanding of the interconnectedness of the various problem parameters to grokking. If the reviewer asks for such a clarification regarding a specific work not covered in our Section 1.1, then we are happy to further discuss it in a revised version.

---

> > ### Author Rebuttal · Reviewer_MztS · 2026-04-02
> >
> > Thank you very much for the comprehensive and careful response. After reading the rebuttal and other reviews, I feel my concerns are mainly addressed. I will update my score accordingly.

---

> > > ### Author Response · Authors · 2026-04-06
> > >
> > > Thank you for carefully reading our rebuttal and for your positive feedback. We are happy to know that you found our response clear and updated your evaluation of our work. Your insightful comments will be very helpful for improving our paper.

---

### Official Review · Reviewer_ARQb · 2026-03-07

**Soundness:** 3
**Presentation:** 4
**Significance:** 4
**Originality:** 3
**Overall Recommendation:** 6
**Confidence:** 3

**Summary:**

This paper investigates grokking, the phenomenon where a model can still generalize, after having overfitted. The grokking effect is analyzed for models operating in a ridge regression problem of learning a teacher function, by training a student model.

Specifically, the authors study grokking both in the case of a zero teacher, and the case where a teacher exists. Theoretically, grokking is proved by upper bounding the largest number of training steps such that the generalization loss is above a certain threshold, and lower bounding the smallest number of training steps for which the generalization loss is below that threshold. The difference between the largest and the smallest number of training steps can be controlled by appropriately fine-tuning the hyper-parameters appearing in each corresponding bound, thereby leading to a control of the grokking effect.

The theoretical results are supported by relevant experiments. The authors provide an extensive experimental setup, followed by a discussion on the findings. Overall, the paper provides new insights on how grokking works in regression settings.

**Compliance With Llm Reviewing Policy:**

Affirmed.

**Final Justification:**

The rebuttal addressed all my concerns, so I see no reason for not increasing my score.

**Key Questions For Authors:**

1. Could you move Sec. 1.1 to an appendix, and leverage the space left to address some more details? For instance, you stress the reasons of "why ridge regression" in the first two pages, but perhaps adding some relevant details in Sec. 2, where ridge regression is formulated, would give a gentle reminder of what's at stake.
2. Could you include a more detailed conclusion in Sec. 6? Right now, it feels a bit disproportional: a lot of space is dedicated to stressing the course of future work (which is fine, of course), with just a concluding remark before that. However, every properly written paper requires an equally properly written ending.

**Limitations:**

Yes.

**Strengths And Weaknesses:**

## Strengths

The paper is well-written and easily readable. All claims are supported by relevant theory, yielding new results and explanations on the grokking phenomenon. The experimental section follows naturally the theoretical findings and appropriately supports them.

Sec. 3 is particularly appealing, where the authors informally explain what their results are about.

## Weaknesses

1. Sec. 1.1 seems a bit odd: you don't need to put additional related work in the main paper (but rather in a relevant appendix), as you deprive it of necessary space, which could be dedicated to adding other details.
2. Very small conclusion; it feels like Sec. 6 is only about future work.

---

> ### Author Rebuttal · Authors · 2026-03-30
>
> We sincerely thank the reviewer for taking the time and effort to review our paper. We appreciate the positive feedback and the encouraging words regarding the significance of our work.
>
> Q1. We thank the reviewer for providing this helpful suggestion on the presentation. We will make use of the additional space given in the camera-ready version to adopt this suggestion, which will further improve the clarity of our paper.
>
> Q2. We also thank the reviewer for suggesting an extended conclusion section that includes further discussion of our contributions in the present work. Similarly to the previous suggestion, we will incorporate this in the final version of the paper.

---

> > ### Author Rebuttal · Reviewer_ARQb · 2026-04-01
> >
> > I thank the authors for addressing my concerns, all of which are covered; to that end, I will also increase my score. Good luck with your future work!

---

> > > ### Author Response · Authors · 2026-04-06
> > >
> > > Thank you for reading our rebuttal carefully and increasing your score. We appreciate your constructive comments throughout the review process, which definitely help us improve the quality of the paper.

---

### Official Review · Reviewer_39zk · 2026-03-11

**Soundness:** 3
**Presentation:** 3
**Significance:** 2
**Originality:** 2
**Overall Recommendation:** 3
**Confidence:** 4

**Summary:**

The paper theoretically discusses the effect of Grokking in a kernel ridge regression setup and extends this theoretical notion experimentally to one-hidden-layer networks with ReLU activations.
Unlike in related works a wide definition of grokking is assumed as the increase of the time gap train and test error pass a certain generalization threshold. Grokking here does not necessarily imply a sudden drop in the generalization loss of the network.
The paper is build up on the three main statements that the system over fits early on; that the temporal gap between the train and test error passing a certain threshold can be made infinitely large with the right choice of hyperparameters and that any such threshold will eventually be undercut by the generalization error.

**Compliance With Llm Reviewing Policy:**

Affirmed.

**Key Questions For Authors:**

1. What happens in the presence of noise both static and dynamic? It is known that this will bound the generalization error in ridge regression. Can their still be made statements on the generalization performance compared to the training error?

2. What does happen for different learning rate, e.g. time-dependent \eta  as that is known to also promote grokking. What happens in the \eta\to0 limit, i.e. gradient flow? Wouldn't one expect the equations to simplify and to allow for a more qualitative understanding?

3. While the paper addresses very well, when the grokking effect get enhanced, can the theory say something on when it gets inhibited?

**Limitations:**

Yes.

**Strengths And Weaknesses:**

Soundness:

The paper seams to be technically sound. All quantities are well defined and all statements are given as theorems with respected proofs in the appendices. The proofs where only spot checked but the methods seems adequate and being applied with appropriate rigor.

Presentation:
The paper is well written, clearly structured and can be easily followed. Notations and assumptions are clearly stated and all the simulations are provided with the corresponding parameters for reproducibility. The paper follows a clear narrative and gives an extensive overview about the concurring literature. All claims the paper made are backed by theoretical results. A potential weakness may be, that the paper may try to overstate its impact on the subject matter as they use a quite loose definition of grokking. Investigating the train-test time lag is, however, also of significance.
In addition we do not find it very useful to address the zero-teacher as its result is fairly trivial.
In general, however, the weaknesses and limitations are discussed in a sincere manner.

Significance:
The paper gives theoretical bounds that may lead interesting insights to hyperparameter tuning. However, it fails to make aid real world setups dealing with grokking which is even discuss in section 6. While in noiseless ridge regression there is a guaranteed convergence, the regularization parameter can be made arbitrarily small to increase the time lag and still converge. With this being a necessity, this generalizes badly to systems with noisy teacher or SGD in cases with non-convex loss landscapes. Therefore the overall significance may be limited and mainly for valuable for theoreticians.

Originality:
As far as we are concerned the tools used are fairly standard calculus tools that have been used in the literature before. In the context of train-test error gaps this results might be novel, theoretical bounds on the test error, however, have been studied excessively in the literature. In case we have over-looked an important novelty we kindly ask the authors to address this.

---

> ### Author Rebuttal · Authors · 2026-03-31
>
> We sincerely thank the reviewer for taking the time and effort to review our paper. We appreciate their positive remarks regarding the rigor, as well as the quality of our presentation. Below, we provide detailed responses to their questions and concerns.
>
> Regarding the significance, the reviewer mentioned that our theory does not generalize to more complicated scenarios such as running SGD on nonconvex losses with noisy data. Apart from experimentally verifying this as detailed below in Q1, we would like to mention that one of the main contributions of this work is to establish the first end-to-end provable grokking result, which we view as a first step towards a more rigorous understanding of grokking, including in more complicated settings as the reviewer suggests. Analyzing the linear case is a natural starting point, and is already highly non-trivial. Moreover, we find that proving the manifestation of grokking in simpler models provides a more surprising result compared to the more complicated settings in which they were first identified (e.g. Power et al. 2022). Lastly, we hope that our results could potentially shed light on understanding grokking in more complex settings and even in modern deep learning.
>
> Regarding the technical tools, while the tools used in the proof are relatively standard, the main challenge in the analysis is to understand how different combinations of hyperparameters affect the three components of grokking simultaneously, and how certain combinations allow for provable grokking. Hence, we believe that our analysis is highly non-trivial.
>
> Q1. Analyzing grokking in the presence of noise is indeed a challenging problem. Our work serves as a starting point for rigorously proving end-to-end grokking, and we agree that considering noisy settings is an interesting direction for future work.
> Since we found the reviewer's question intriguing, we conducted additional experiments on our ridge regression model with label noise, and we are glad to report that we did observe grokking when the labels are contaminated with mean-zero variance-$0.01$ Gaussian noise, a finding that we will add in a revised version of the paper. We believe that establishing rigorous provable grokking for this setup is intriguing and challenging, and we intend to study this in the future.
>
> Q2. When the learning rate decreases as a function of the training time, grokking will be enhanced (to observe longer generalization delay). Following the reviewer's question, we empirically studied the effects of learning rate schedules on the grokking time, and we found that this indeed amplifies grokking. Following this empirical finding, we also believe that our analysis can provide quantitative bounds for the grokking time in such a setting.
> We also posit that our general approach can be used to obtain quantitative bounds for gradient flow. Note that we can scale down the step size $\eta$ and analyze the resulting effects on our derived bounds in Theorems 4.4--4.6. Since the functional dependence on $\eta$ typically takes the general form $(1-\Theta(\eta))^t$, which for sufficiently large $\eta\cdot t$ behaves like $\Theta(\exp(-\eta t))$, we have that decreasing the step size $\eta$ and increasing the number of GD iterations $t$ by the same multiplicative factor will barely change the bounds derived in our analysis. Thus, we expect to see the same grokking behavior when considering GF instead of GD. We thank the reviewer for this interesting viewpoint that broadens the impact of our analysis, and we will add this discussion in a revised version of our paper.
>
> Q3. Our theory provides a rigorous characterization of how model hyperparameters affect grokking, and thus provides guidance for how to enhance as well as inhibit grokking. As a concrete example, when observing grokking, increasing the weight decay would help to eliminate grokking.

---

> > ### Author Rebuttal · Reviewer_39zk · 2026-04-03
> >
> > I thank the authors for thoroughly answering my questions and addressing my concerns. While I still think that the impact of the work might be only of limited scope, I highly value the level of mathematical rigor and the clear presentation and will raise my score accordingly.
> >
> > Score: Soundness and Presentation raised to 4 points, Originality and Significance to 3 points. Overall score raised to 5 points.

---

> > > ### Author Response · Authors · 2026-04-06
> > >
> > > Thank you very much for carefully reading our rebuttal and for your encouraging feedback. We are grateful that you found our response helpful. Your insightful comments and novel questions are very conducive to us for improving the paper. We would like to gently remind you to update your score before the deadline.

---

### Decision · Program_Chairs · 2026-04-30

**Decision:**

Accept (spotlight)

**Comment:**

Summary: This paper provides a theoretical analysis of grokking in overparameterized ridge regression trained with gradient descent and weight decay, establishing end-to-end provable bounds on the time lag between training and test error crossing a given threshold. The analysis is extended experimentally to single-hidden-layer ReLU networks.

Reviewer Consensus and Concerns: All four reviewers acknowledged the paper's technical rigor and clarity of presentation. The shared concern across reviewers was the limited scope and practical significance of the results: the theory applies only to linear (kernel) models with weight decay, uses a relaxed definition of grokking (time lag rather than sharp phase transition with plateau), and it remains unclear how well the insights transfer to nonlinear deep networks where grokking is most interesting. Reviewers also questioned robustness to noise, dependence on weight decay, and whether the observed phenomenon truly captures the essence of grokking versus simply demonstrating a train-test lag in linear models.

Assessment of the Rebuttal: The authors provided a thorough and effective rebuttal. They ran additional experiments (noisy labels, random Fourier features, learning rate schedules), clarified the plateau behavior through an alternative accuracy-like metric, and engaged substantively with conceptual questions (GP interpretation, connection to feature learning). All four reviewers explicitly confirmed their concerns were fully resolved and three raised their scores accordingly.

Overall Assessment: Post-rebuttal scores are 5, 6, 4, 6 (mean ~5.25). The paper's main strength is that it is the first end-to-end provable result on grokking, with clean quantitative bounds and an honest, well-written presentation. Its main weakness is that the setting is restrictive to linear models, weight decay required, no noise in the theory, and the definition of grokking is broader than what the community typically finds surprising. The practical implications remain limited. That said, the paper establishes a solid theoretical foundation that future work can build upon, and the rebuttal convincingly addressed all raised concerns.

Recommendation: I lean toward  strong accept. While the scope is narrow, the paper makes a technically sound and novel contribution to an important phenomenon, the presentation is strong, and the reviewers were uniformly satisfied after discussion. The work fills a clear gap in the literature as a rigorous first step toward understanding grokking theoretically.